# FreCaS: Efficient Higher-Resolution Image Generation via Frequency-aware Cascaded Sampling

**Zhengqiang Zhang**[1,2]**, Ruihuang Li**[1]**, Lei Zhang**[1,2,†]
[1]The Hong Kong Polytechnic University      [2]OPPO Research Institute
zhengqiang.zhang@connect.polyu.hk, cslzhang@comp.polyu.edu.hk
[†]Corresponding author

## Abstract

While image generation with diffusion models has achieved a great success, generating images of higher resolution than the training size remains a challenging task due to the high computational cost. Current methods typically perform the entire sampling process at full resolution and process all frequency components simultaneously, contradicting with the inherent coarse-to-fine nature of latent diffusion models and wasting computations on processing premature high-frequency details at early diffusion stages. To address this issue, we introduce an efficient **Fre**quency-aware **Ca**scaded **S**ampling framework, **FreCaS** in short, for higher-resolution image generation. FreCaS decomposes the sampling process into cascaded stages with gradually increased resolutions, progressively expanding frequency bands and refining the corresponding details. We propose an innovative frequency-aware classifier-free guidance (FA-CFG) strategy to assign different guidance strengths for different frequency components, directing the diffusion model to add new details in the expanded frequency domain of each stage. Additionally, we fuse the cross-attention maps of previous and current stages to avoid synthesizing unfaithful layouts. Experiments demonstrate that FreCaS significantly outperforms state-of-the-art methods in image quality and generation speed. In particular, FreCaS is about $2.86\times$ and $6.07\times$ faster than ScaleCrafter and DemoFusion in generating a $2048\times2048$ image using a pre-trained SDXL model and achieves an $\text{FID}_b$ improvement of 11.6 and 3.7, respectively. FreCaS can be easily extended to more complex models such as SD3. The source code of FreCaS can be found at https://github.com/xtudbxk/FreCaS.

## 1 Introduction

In recent years, diffusion models, such as Imagen (Saharia et al., 2022), SDXL (Podell et al., 2023), PixelArt-$\alpha$ (Chen et al., 2023) and SD3 Esser et al. (2024), have achieved a remarkable success in generating high-quality natural images. However, these models face challenges in generating very high resolution images due to the increased complexity in high-dimensional space. Though efficient diffusion models, including ADM (Dhariwal & Nichol, 2021), CascadedDM (Ho et al., 2022) and LDM (Rombach et al., 2022), have been developed, the computational burden of training diffusion models from scratch for high-resolution image generation remains substantial. As a result, popular diffusion models, such as SDXL (Podell et al., 2023) and SD3 (Esser et al., 2024), primarily focus on generating $1024 \times 1024$ resolution images. It is thus increasingly attractive to explore training-free strategies for generating images at higher resolutions, such as $2048 \times 2048$ and $4096 \times 4096$, using pre-trained diffusion models.

MultiDiffusion (Bar-Tal et al., 2023) is among the first works to synthesize higher-resolution images using pre-trained diffusion models. However, it suffers from issues such as object duplication, which largely reduces the image quality. To address these issues, Jin et al. (2024) proposed to manually adjust the scale of entropy in the attention operations. He et al. (2023) and Huang et al. (2024) attempted to enlarge the receptive field by replacing the original convolutional layers with strided ones, while Zhang et al. (2023) explicitly resizes the intermediate feature maps to match the train-

ing size. Du et al. (2024) and Lin et al. (2024) took a different strategy by generating a reference image at the base resolution and then using it to guide the whole sampling process at higher resolutions. Despite the great advancements, these methods still suffer from significant inference latency, hindering their broader applications in real world.

In this paper, we propose an efficient **Fre**quency-aware **Ca**scaded **S**ampling framework, namely **FreCaS**, for training-free higher-resolution image generation. Our proposed FreCaS framework is based on the observation that latent diffusion models exhibit a coarse-to-fine generation manner in the frequency domain. In other words, they first generate low-frequency contents in early diffusion stages and gradually generate higher-frequency details in later stages. Leveraging this insight, we generate higher-resolution images through multiple stages of increased resolutions, progressively synthesizing details of increased frequencies. FreCaS avoids unnecessary computations during the early diffusion stages as high-frequency details are not yet required.

In the latent space, the image representation expands its frequency range as the resolution increases. To encourage detail generation within the expanded frequency band, we introduce a novel frequency-aware classifier-free guidance (FA-CFG) strategy, which prioritizes newly introduced frequency components by assigning them higher guidance strengths in the sampling process. Specifically, we decompose both unconditional and conditional denoising scores into two parts: low-frequency component, which captures content from earlier stages, and high-frequency component, which corresponds to the newly increased frequency band. FA-CFG applies the classifier-free guidance to different frequency components with different strengths, and outputs the final denoising score by combining the adjusted components. The FA-CFG strategy can synthesize much clear details while maintaining computational efficiency. Additionally, to alleviate the issue of unfaithful layouts, such as duplicated objects mentioned in Jin et al. (2024), we reuse the cross-attention maps (CA-maps) from the previous stage, which helps maintaining consistency in image structure across different stages and ensuring more faithful object representations.

In summary, our main contributions are as follows:

- We propose FreCaS, an efficient frequency-aware cascaded sampling framework for training-free higher-resolution image generation. FreCaS leverages the coarse-to-fine nature of the latent diffusion process, thereby reducing unnecessary computations associated with processing premature high-frequency details.
- We design a novel FA-CFG strategy, which assigns different guidance strengths to components of different frequencies. This strategy enables FreCaS to focus on generating contents of newly introduced frequencies in each stage, and hence synthesize clearer details. In addition, we fuse the CA-maps of previous stage and current stage to maintain a consistent image layouts across stages.
- We demonstrate the efficiency and effectiveness of FreCaS through extensive experiments conducted on various pretrained diffusion models, including SD2.1, SDXL and SD3, validating its broad applicability and versatility.

## 2 RELATED WORKS

### 2.1 DIFFUSION MODELS

Diffusion models have gained significant attentions due to their abilities to generate high-quality natural images. Ho et al. (2020) pioneered the use of a variance-preserving diffusion process to bridge the gap from natural images to pure noises. Dhariwal & Nichol (2021) exploited various network architectures and achieved superior image quality than contemporaneous GAN models. Ho & Salimans (2022) introduced a novel classifier-free guidance strategy that attains both generated image quality and diversity. However, the substantial model complexity makes high-resolution image synthesis challenging. Ho et al. (2022) proposed a novel cascaded framework that progressively increases image resolutions. Rombach et al. (2022) performed the diffusion process in the latent space of a pre-trained autoencoder, enabling high-resolution image synthesis with reduced computational cost. (Esser et al., 2024) presented SD3, which employs the rectified flow matching (Lipman et al., 2022; Liu et al., 2022) at the latent space and demonstrates superior performance. Despite the great progress, it still requires substantial efforts to train a high-resolution diffusion model from scratch. Therefore, training-free higher-resolution image synthesis attracts increasing attentions.

## 2.2 TRAINING-FREE HIGHER-RESOLUTION IMAGE SYNTHESIS

A few methods have been developed to leverage pre-trained diffusion models to generate images of higher resolutions than the training size. MultiDiffusion (Bar-Tal et al., 2023) is among the first methods to bind multiple diffusion processes into one unified framework and generates seamless higher-resolution images. However, the results exhibit unreasonable image structures such as duplicated objects. AttnEntropy (Jin et al., 2024) alleviates this problem by re-normalizing the entropy of attention blocks during sampling. On the other hand, ScaleCrafter (He et al., 2023) and FouriScale (Huang et al., 2024) expand the receptive fields of pre-trained networks to match higher inference resolutions, thereby demonstrating improved image quality. HiDiffusion (Zhang et al., 2023) dynamically adjusts the feature sizes to match the training dimensions. DemoDiffusion (Du et al., 2024) and AccDiffusion (Lin et al., 2024) first generate a reference image at standard resolutions and then use this image to guide the generation of images at higher resolutions. Despite their success, the above mentioned approaches neglect the coarse-to-fine nature of image generation and generate image contents of all frequencies simultaneously, resulting in long inference latency and limiting their broader applications.

To address this issue, we propose an efficient FreCaS framework for training-free higher-resolution image synthesis. FreCaS divides the entire sampling process into stages of increasing resolutions, gradually synthesizing components of different frequency bands, thereby reducing the unnecessary computation of handling premature high-frequency details in early sampling stages. It is worth noting that DemoFusion (Du et al., 2024) and ResMaster (Shi et al., 2024) also employ a cascaded sampling scheme. However, there exist fundamental differences between FreCaS and them: DemoFusion and ResMaster perform a complete diffusion process at each resolution, whereas FreCaS transitions the diffusion from low to high resolutions in just one process. This distinction makes our method significantly more efficient than them while achieving better image quality.

## 3 METHOD

This section presents the details of the proposed FreCaS framework, which leverages the coarse-to-fine nature of latent diffusion models and constructs a frequency-aware cascaded sampling strategy to progressively refine high-frequency details. We first introduce the notations and concepts that form the basis of our approach (see Section 3.1). Then, we delve into the key components of our method: FreCaS framework (see Section 3.2), FA-CFG strategy (see Section 3.3), and CA-maps re-utilization (see Section 3.4).

### 3.1 PRELIMINARIES

**Diffusion models.** Diffusion models (Ho et al., 2020; Dhariwal & Nichol, 2021) transform complex image distributions into the Gaussian distribution, and vice versa. They gradually inject Gaussian noises into the image samples, and then use a reverse process to remove noises from them, achieving image generation. Most recent diffusion models operate in the latent space and utilize a discrete timestep sampling process to synthesize images. Specifically, for a $T$-step sampling process, a latent noise $z_T$ is drawn from a standard Gaussian distribution, and then iteratively refined through a few denoising steps until converged to the clean signal latent $z_0$. Finally, the natural image $x$ is decoded from $z_0$ using a decoder $\mathcal{D}$. The whole process can be written as follows:

$$z_T \sim \mathcal{N}(\mathbf{0}, \mathbf{I}) \rightarrow z_{T-1} \rightarrow \cdots \rightarrow z_1 \rightarrow z_0 \rightarrow x = \mathcal{D}(z_0). \tag{1}$$

For each denoising step, current works typically adopt the classifier-free guidance (CFG) (Ho & Salimans, 2022) to improve image quality. It predicts an unconditional denoising score $\boldsymbol{\epsilon}_{unc}$ and a conditional denoising score $\boldsymbol{\epsilon}_c$. The final denoising score is obtained via a simple extra-interpolation process as $\hat{\boldsymbol{\epsilon}} = (1 - w) \cdot \boldsymbol{\epsilon}_{unc} + w \cdot \boldsymbol{\epsilon}_c$, where $w$ denotes the guidance strength.

**Resolution and frequency range.** The resolution of a latent $z$ determines its sampling frequency (Rissanen et al., 2023), thereby influencing its frequency domain characteristics. Specifically, if a latent of unit length has a resolution of $s \times s$, its sampling frequency $f_s$ can be defined as the number of samples per unit length, which is $s$. The Nyquist frequency is then obtained as $\frac{f_s}{2} = \frac{s}{2}$. Therefore, the frequency of the latent $z$ ranges from $[0, \frac{s}{2}]$. Reducing its resolution to

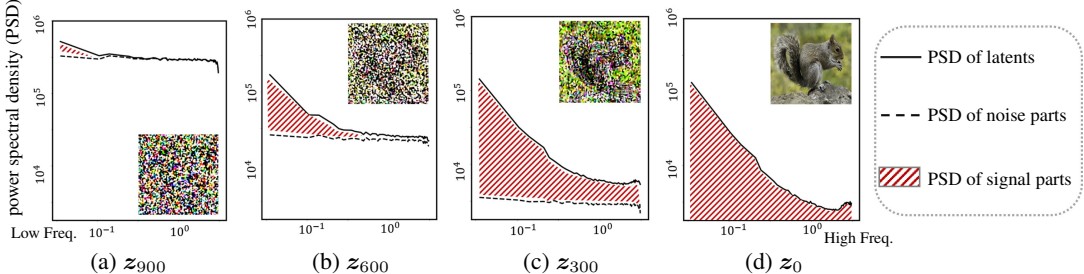

Figure 1: From (a) to (d), the sub-figures show the PSD curves of latents $z_{900}$, $z_{600}$, $z_{300}$ and $z_0$ of SDXL, respectively. One can see that the energy of synthesized clean signals (the red slashed regions) first emerges in the low-frequency band and gradually expands to high-frequency band.

$s_l \times s_l$ narrows the frequency range to $[0, \frac{s_l}{2}]$. As a result, higher resolutions capture a broader frequency domain, while lower resolutions lead to a narrower frequency spectrum.

### 3.2 FREQUENCY-AWARE CASCADED SAMPLING

Pixel space diffusion models exhibit a coarse-to-fine behavior in the image synthesis process (Rissanen et al., 2023; Teng et al., 2024). In this section, we show that such a behavior is also exhibited for latent diffusion models during the sampling process, which inspires us to develop a frequency-aware cascaded sampling framework for generating higher-resolution images.

**PSD curves in latent space.** The power spectral density (PSD) is a powerful tool for analyzing the energy distribution of signals along the frequency spectrum. Rissanen et al. (2023) and Teng et al. (2024) have utilized PSD to study the behaviour of intermediate states in the pixel diffusion process. Here, we compute the PSD of the latent signals over a collection of 100 natural images using the pre-trained SDXL model (Podell et al., 2023). Figure 1 shows the PSD curves of $z_{900}$, $z_{600}$, $z_{300}$ and $z_0$. The solid line denotes the PSD curve of intermediate noise corrupted latent, while the dashed line represents the PSD of Gaussian noise corrupted into the latent. The inner area between the two curves (marked with red slashes) indicates the energy of clean signal latent being synthesized. One can see that the clean image signals emerge from the low-frequency band (see $z_{900}$ and $z_{600}$) and gradually expand to the high-frequency band (see $z_{300}$ and $z_0$) during the sampling process. These observations confirm the coarse-to-fine nature of image synthesis in the latent diffusion process, where low-frequency content is generated first, followed by high-frequency details.

**Framework of FreCaS.** Based on the above observation, we developed an efficient FreCaS framework to progressively generate image contents of higher frequency bands, reducing unnecessary computations in processing premature high-frequency details in early diffusion stages. As shown in Figure 2(a), our FreCaS divides the entire $T$-step sampling process into $N+1$ stages of increasing resolutions. The initial stage performs the sampling process at the default training size $s_0$ with a frequency range of $[0, \frac{s_0}{2}]$. Each of the subsequent stages increases the sampling size to its predecessor, gradually expanding the frequency domain. At the final stage, the latent reaches the target resolution $s_N$, achieving a full frequency range from 0 to $\frac{s_N}{2}$.

Specifically, we begin with a pure noise latent $z_T^{s_0}$ at stage $s_0$, and iteratively perform reverse sampling until obtaining the last latent in this stage, denoted by $z_L^{s_0}$. Next, we transition $z_L^{s_0}$ to the first latent, denoted by $z_F^{s_1}$, in next stage, as illustrated by the blue dashed arrow in Figure 2(a). This procedure is repeated until the latent feature reaches the target size, resulting in $z_0^{s_N}$. The final image $x$ is obtained by applying the decoder to $z_0^{s_N}$ so that $x = \mathcal{D}(z_0^{s_N})$. With such a sampling pipeline, FreCaS ensures a gradual refinement of details across coarse-to-fine scales, ultimately producing a high-quality and high-resolution image with minimum computations.

For the transition between two adjacent stages, we perform five steps to convert the last latent of previous stage $z_L^{s_{i-1}}$ to the first latent of next stage $z_F^{s_i}$:

$$z_L^{s_{i-1}} \xrightarrow{\text{denoise}} \hat{z}_0^{s_{i-1}} \xrightarrow{\text{decode}} \hat{x}^{s_{i-1}} \xrightarrow{\text{interpolate}} \hat{x}^{s_i} \xrightarrow{\text{encode}} z_0^{s_i} \xrightarrow{\text{diffuse}} z_F^{s_i}, \tag{2}$$

where "denoise" and "diffuse" are standard diffusion operations, "decode and "encode" are performed using the decoder and encoder, respectively, and "interpolation" adjusts the resolutions using

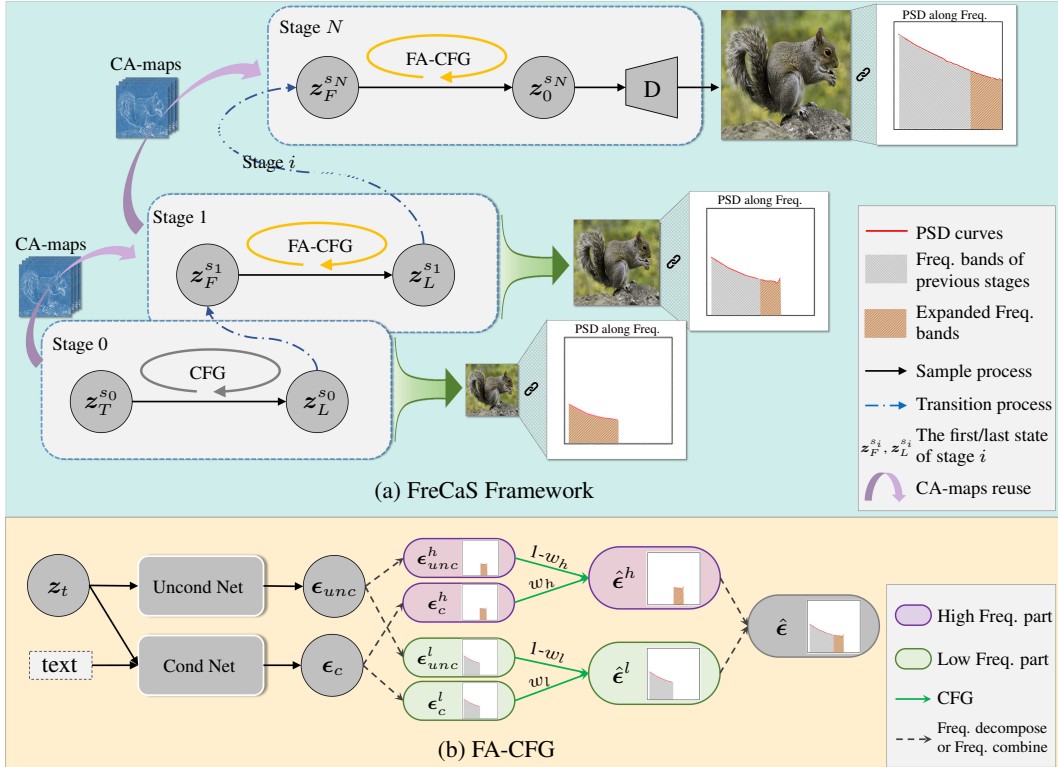

Figure 2: (a) The overall framework of FreCaS. The entire $T$-step sampling process is divided into $N + 1$ stages of increasing resolutions and expanding frequency bands. FreCaS starts the sampling process at the training size and obtains the last latent $z_L^{s_0}$ at that stage. Then, FreCaS continues the sampling from the first latent $z_F^{s_1}$ at the next stage with a larger resolution and expanded frequency domain. This procedure is repeated until the final latent $z_0^{s_N}$ at stage $N$ is obtained. A decoder is then used to generate the final image. (b) FA-CFG strategy. We separate the original denoising scores into low-frequency and high-frequency components and assign a higher CFG strength to the high-frequency part. The two parts are then combined to obtain the final denoising score $\hat{\epsilon}$.

the bilinear interpolation. To determine the timestep of $z_F^{s_i}$, we follow previous works (Hoogeboom et al., 2023; Chen, 2023; Gu et al., 2023; Teng et al., 2024) to keep the signal-to-noise ratio (SNR) equivalence between $z_L^{s_{i-1}}$ and $z_F^{s_i}$. Please refer to Appendix A for more details.

## 3.3 FA-CFG STRATEGY

Our FreCaS framework progressively transitions the latents to stages with higher resolutions and extended high-frequency bands. To ensure that the diffusion models focus more on generating contents of newly introduced frequencies, we propose a novel FA-CFG strategy, which assigns higher guidance strength to the new frequency components. In FreCaS, upon transitioning to stage $s_i$, the latent increases its resolution from $s_{i-1}$ to the higher resolution $s_i$, thereby expanding the frequency band from $[0, \frac{s_{i-1}}{2}]$ to $[0, \frac{s_i}{2}]$. This inspires us to divide the latents into two components: a low-frequency component ranging from $[0, \frac{s_{i-1}}{2}]$ and a high-frequency component covering the frequency interval $(\frac{s_{i-1}}{2}, \frac{s_i}{2}]$. The former preserves the generated contents from previous stages, whereas the latter is reserved for the contents to be generated in this stage. Our goal is to encourage the diffusion models to generate natural details and textures in the newly expanded frequency band.

To achieve the above mentioned goal, we propose to perform CFG on the two frequency-aware parts with different guidance strengths. The entire process is illustrated in Figure 2(b). First, we obtain the unconditional denoising score $\epsilon_{unc}$ and conditional denoising score $\epsilon_c$ using the pre-trained diffusion network. Then, we split the scores into a low-frequency part and a high-frequency part. The former is extracted by downsampling the scores and then resizing them back, while the latter is the residual by subtracting the low-frequency part from the original denoising scores. Subsequently, we apply the CFG strategy to the two parts with different weights. Specifically, for the low-frequency

part, we assign the normal guidance strength $w_l$, while for the high-frequency part, we use a much higher weight $w_h$ to prioritize content generation in this frequency band. The final denoising score is obtained by summing up the two parts. This process can be expressed as:

$$\hat{\boldsymbol{\epsilon}} = \hat{\boldsymbol{\epsilon}}^l + \hat{\boldsymbol{\epsilon}}^h = (1 - w_l) \cdot \boldsymbol{\epsilon}_{unc}^l + w_l \cdot \boldsymbol{\epsilon}_c^l + (1 - w_h) \cdot \boldsymbol{\epsilon}_{unc}^h + w_h \cdot \boldsymbol{\epsilon}_c^h, \tag{3}$$

where $\hat{\boldsymbol{\epsilon}}^l$ and $\hat{\boldsymbol{\epsilon}}^h$ are the low-frequency and high-frequency parts of $\hat{\epsilon}$, respectively. Similarly, $\boldsymbol{\epsilon}_{unc}^l$, $\boldsymbol{\epsilon}_{unc}^h$, $\boldsymbol{\epsilon}_c^l$ and $\boldsymbol{\epsilon}_c^h$ follow the same notation.

### 3.4 CA-MAPS REUTILIZATION

When applied to higher resolutions, pre-trained diffusion models often present unreasonable image structures, such as duplicated objects. To address this issue, we propose to reuse the CA-maps from the previous stage to maintain layout consistency across stages. The CA-maps represent attention weights from cross-attention interactions between spatial features and textual embeddings, effectively capturing the semantic layout of the generated images. Specifically, we average the CA-maps of all cross-attention blocks when predicting $\boldsymbol{z}_L^{s_{i-1}}$ at stage $s_{i-1}$. After transitioning to stage $s_i$, we replace the current CA-maps of each cross-attention block using its linear interpolation with the averaged CA-maps $\overline{M}_L^{s_{i-1}}$ as follows:

$$M_t^{s_i} = (1 - w_c) \cdot M_t^{s_i} + w_c \cdot \overline{M}_L^{s_{i-1}}, \tag{4}$$

where $M_t^{s_i}$ is the CA-maps at step $t$ of stage $s_i$. In this way, FreCaS can effectively maintain content consistency and prevent unexpected objects or textures during higher-resolution image generation.

## 4 EXPERIMENTS

### 4.1 EXPERIMENTAL SETTINGS

**Implementation details.** We evaluate FreCaS on three widely-used pre-trained diffusion models: SD2.1 (Rombach et al., 2022), SDXL (Podell et al., 2023) and SD3 (Esser et al., 2024). The sizes of generated images are $\times 4$ and $\times 16$ the original training size. Specifically, we generate images of $1024 \times 1024$ and $2048 \times 2048$ for SD2.1, while $2048 \times 2048$ and $4096 \times 4096$ for SDXL. For SD3, we only generate images of $2048 \times 2048$ due to the GPU memory limitation. We randomly select 10K, 5K, and 1K prompts from the LAION5B aesthetic subset for generating images of $1024 \times 1024$, $2048 \times 2048$, and $4096 \times 4096$, respectively. We follow the default settings and perform a 50-step sampling process with DDIM sampler for SD2.1 and SDXL, and perform a 28-step sampling process with a flow matching based Euler solver for SD3. For $\times 4$ experiments, we employ two sampling stages at the training size and target size, respectively. For $\times 16$ experiments, we employ three sampling stages at the training size, $4 \times$ training size and $16 \times$ training size, respectively. More details can be found in Appendix B.

**Evaluation metrics.** We employ the Fréchet Inception Distance (FID) (Heusel et al., 2017) and Inception Score (IS) (Salimans et al., 2016) to measure the quality of generated images. Following He et al. (2023), we also employ $FID_b$ as the metric, which is computed on the samples of training size and target size. As suggested by Du et al. (2024), we report $FID_p$ and $IS_p$, which compute the metrics at patch level, to better evaluate the image details. The CLIP score (Radford et al., 2021) is utilized to measure the text prompt alignment of generated images. As in previous works (Zhang et al., 2023), we measure the model latency on a single NVIDIA A100 GPU with a batch size of 1. We generate five images and report the averaged latency of the last three images for all methods. Moreover, we conduct a user study and employ the non-reference image quality assessment metrics to further evaluate our FreCaS. Please refer to Appendix C for the details.

### 4.2 EXPERIMENTS ON SD2.1 AND SDXL

For experiments on SD2.1, we compare FreCaS with DirectInference, MultiDiffusion (Bar-Tal et al., 2023), AttnEntropy (Jin et al., 2024), ScaleCrafter (He et al., 2023), FouriScale (Huang et al., 2024) and HiDiffusion (Zhang et al., 2023). For experiments on SDXL, we compare with DirectInference, AttnEntropy, ScaleCrafter, FouriScale, HiDiffusion, AccDiffusion (Lin et al., 2024) and DemoFusion (Du et al., 2024). We further compare our FreCaS with training-based methods (Pixart-Sigma (Chen et al., 2024) and UltraPixel (Ren et al., 2024)) and super-resolution methods (ESRGAN (Wang et al., 2021) and SUPIR (Yu et al., 2024)) in Appendix D.

Table 1: Experiments on $\times 4$ and $\times 16$ generation of SD2.1 and SDXL. "DO" means "duplicated object", which indicates whether the method takes the duplicated object problem into consideration. "SpeedUP" denotes the efficiency speed-up over the DirectInference baseline. The **red** and blue indicate the best and second ones among all methods that consider the duplicated object problem.

| | | Methods | DO | FID | $FID_b\downarrow$ | $FID_p\downarrow$ | IS↑ | $IS_p\uparrow$ | CLIP SCORE↑ | Latency(s)↓ | SpeedUP↑ |
|---|---|---|---|---|---|---|---|---|---|---|---|
| SD2.1 | ×4 | DirectInference | ✗ | 31.07 | 34.54 | 23.84 | 15.00 | 17.26 | 32.01 | 5.50 | 1x |
| | | MultiDiffusion | ✗ | 21.05 | 22.44 | 14.68 | 17.46 | 18.29 | 32.49 | 120.21 | 0.046× |
| | | AttnEntropy | ✓ | 28.33 | 30.63 | 21.34 | 15.67 | 17.71 | 32.28 | 5.56 | 0.99× |
| | | ScaleCrafter | ✓ | 16.65 | 13.18 | 22.44 | 17.42 | 16.29 | 32.88 | 6.36 | 0.86× |
| | | FouriScale | ✓ | 19.01 | 15.33 | 23.26 | 17.11 | 15.57 | 32.92 | 11.06 | 0.50× |
| | | HiDiffusion | ✓ | 19.95 | 16.21 | 25.26 | 17.13 | 16.12 | 32.37 | 3.57 | 1.54× |
| | | **Ours** | ✓ | 16.38 | 13.14 | 21.23 | 17.55 | 16.04 | 32.33 | 2.56 | 2.16× |
| | ×16 | DirectInference | ✗ | 124.5 | 128.3 | 50.23 | 8.84 | 15.30 | 27.67 | 49.27 | 1× |
| | | MultiDiffusion | ✗ | 67.44 | 74.15 | 15.28 | 8.75 | 18.82 | 31.14 | 926.33 | 0.05× |
| | | AttnEntropy | ✓ | 122.6 | 127.6 | 46.52 | 9.31 | 16.25 | 28.33 | 49.33 | 1.00× |
| | | ScaleCrafter | ✓ | 34.47 | 34.55 | 57.47 | 13.02 | 12.12 | 31.44 | 92.86 | 0.53× |
| | | FouriScale | ✓ | 34.17 | 34.13 | 58.01 | 12.79 | 13.15 | 31.68 | 90.13 | 0.55× |
| | | HiDiffusion | ✓ | 33.15 | 34.17 | 70.58 | 13.49 | 11.87 | 31.09 | 18.22 | 2.70× |
| | | **Ours** | ✓ | 19.95 | 20.11 | 43.71 | 15.22 | 13.74 | 31.92 | 13.35 | 3.69× |
| SDXL | ×4 | DirectInference | ✗ | 39.15 | 43.83 | 29.71 | 11.52 | 14.60 | 32.51 | 34.10 | 1× |
| | | AttnEntropy | ✓ | 36.54 | 41.30 | 27.67 | 11.69 | 15.04 | 32.71 | 34.36 | 0.99× |
| | | ScaleCrafter | ✓ | 22.76 | 24.23 | 23.17 | 14.10 | 14.97 | 32.70 | 39.64 | 0.86× |
| | | FouriScale | ✓ | 26.44 | 26.88 | 27.24 | 13.97 | 14.44 | 32.90 | 66.18 | 0.52× |
| | | HiDiffusion | ✓ | 21.67 | 20.69 | 21.80 | 15.56 | 15.93 | 32.62 | 18.38 | 1.86× |
| | | AccDiffusion | ✓ | 19.87 | 17.62 | 21.11 | 17.07 | 16.15 | 32.66 | 102.46 | 0.33× |
| | | DemoFusion | ✓ | 18.77 | 16.33 | 18.77 | 17.10 | 17.21 | 33.16 | 83.95 | 0.41× |
| | | **Ours** | ✓ | 16.48 | 12.63 | 17.91 | 17.18 | 17.31 | 33.28 | 13.84 | 2.46× |
| | ×16 | DirectInference | ✗ | 145.4 | 151.3 | 62.39 | 6.41 | 11.66 | 28.24 | 312.36 | 1× |
| | | AttnEntropy | ✓ | 142.1 | 148.9 | 60.54 | 6.46 | 12.44 | 28.46 | 312.46 | 1.00× |
| | | ScaleCrafter | ✓ | 71.49 | 75.11 | 73.21 | 8.68 | 9.81 | 30.76 | 560.91 | 0.56× |
| | | FouriScale | ✓ | 98.01 | 77.63 | 84.05 | 8.00 | 9.41 | 30.78 | 534.08 | 0.58× |
| | | HiDiffusion | ✓ | 81.48 | 83.41 | 120.1 | 9.79 | 9.56 | 29.18 | 101.59 | 3.07× |
| | | AccDiffusion | ✓ | 50.47 | 48.15 | 46.07 | 12.11 | 11.75 | 32.26 | 763.23 | 0.41× |
| | | DemoFusion | ✓ | 47.80 | 44.54 | 35.52 | 12.38 | 13.82 | 33.03 | 649.25 | 0.48× |
| | | **Ours** | ✓ | 42.75 | 40.63 | 39.82 | 12.68 | 14.16 | 33.03 | 85.87 | 3.64× |

**Quantitative results.** Table 1 presents quantitative comparisons for $\times 4$ and $\times 16$ generation between FreCaS and its competitors. We can see that FreCaS not only outperforms other methods on synthesized image quality but also exhibits significantly faster inference speed. In specific, FreCaS achieves the best FID scores in all experiments of SD2.1 and SDXL, achieving clear advantages over the other methods. In terms of the IS metric, FreCaS performs the best in most cases, only slightly lagging behind DemoFusion on the $\times 16$ experiments of SDXL. (Note that DirectInference and MultiDiffusion occasionally achieve higher $FID_p$ and $IS_p$ scores because they disregard the issue of duplicated objects.) For CLIP score, FreCaS obtains the best results on 3 out of the 4 cases, except for the less challenging $\times 4$ generation with SD2.1.

While having superior image quality metrics, FreCaS demonstrates impressive efficiency. It shows more than $2\times$ speedup over DirectInference on $\times 4$ generation experiments, and shows more than $3.6\times$ speedup on the $\times 16$ generation experiments. DemoFusion, which is overall the second best method in terms of image quality, is significantly slower than FreCaS. Its latency is about $6\times$ and $7.5\times$ longer than FreCas on $\times 4$ and $\times 16$ experiments, respectively. On the other hand, HiDiffusion, which is the second faster method, sacrifices image quality for speed. For example, on the $\times 16$ experiment with SD2.1, HiDiffusion achieves a latency of 18.22s but its $FID_b$ score is 34.17. In contrast, FreCaS is faster (13.35s) and has a much better $FID_b$ score (20.11).

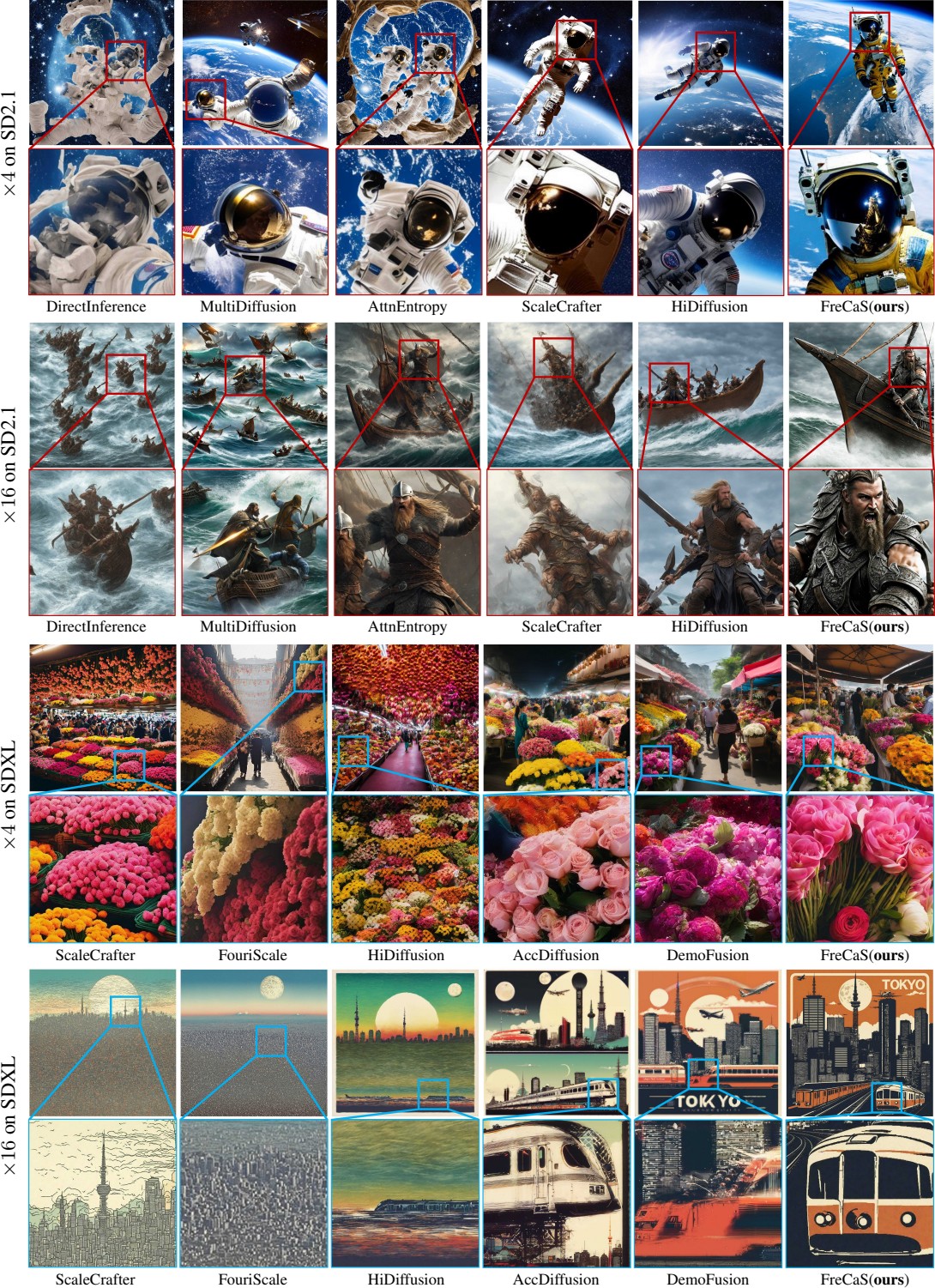

Figure 3: Visual comparison on ×4 and ×16 experiments of SD2.1 and SDXL. From top to bottom, the prompts used in the four groups of examples are: 1. "A cosmic traveler, floating in zero gravity, spacesuit reflecting the Earth below, stars twinkling in the distance." 2. "A fierce Viking, axe in hand, leading a raid, the longship slicing through the waves." 3. "A bustling flower market, stalls filled with bouquets, the air thick with fragrance, people selecting their favorites." 4. "Tokyo Japan Retro Skyline, Airplane, Railroad Train, Moon etc. - Modern Postcard". Zoom-in for better view.

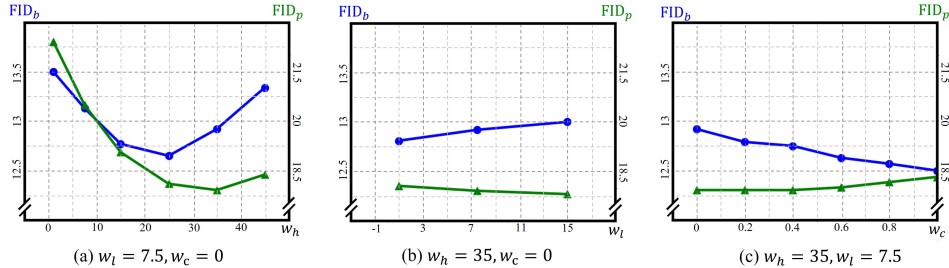

(a) $w_l = 7.5, w_c = 0$      (b) $w_h = 35, w_c = 0$      (c) $w_h = 35, w_l = 7.5$

Figure 4: Ablation studies on $w_l$ and $w_h$ in FA-CFG strategy and $w_c$ in CA-maps reutilization.

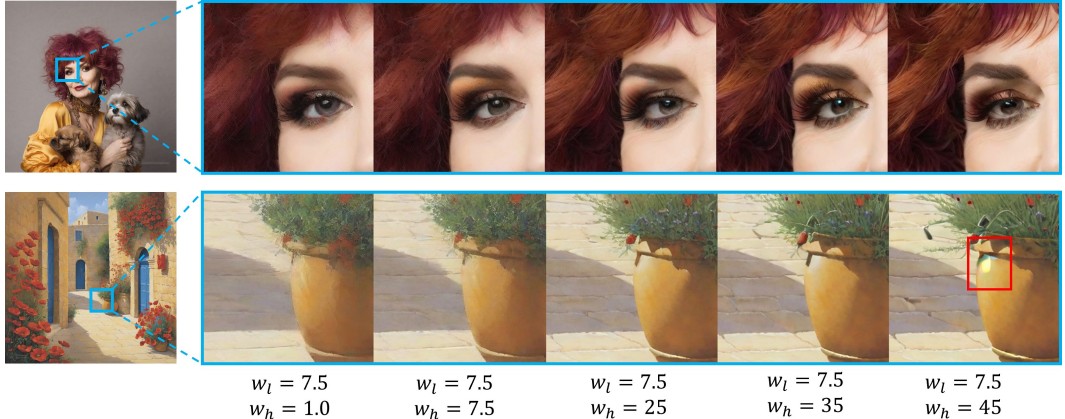

| $w_l = 7.5$ | $w_l = 7.5$ | $w_l = 7.5$ | $w_l = 7.5$ | $w_l = 7.5$ |
| $w_h = 1.0$ | $w_h = 7.5$ | $w_h = 25$ | $w_h = 35$ | $w_h = 45$ |

Figure 5: Visual results of adjusting $w_h$ in the FA-CFG strategy. From top to bottom, the prompts are "Eccentric Shaggy Woman with Pet - Little Puppy" and "Rabat Painting - Mdina Poppies Malta by Richard Harpum", respectively.

**Qualitative results.** Figure 3 illustrates visual comparisons between FreCaS and competitive approaches. From top to bottom are four groups of examples, presenting the results of $\times 4$ generation of SD2.1, $\times 16$ generation of SD2.1, $\times 4$ generation of SDXL, and $\times 16$ generation of SDXL, respectively. In each group, the top row shows the generated images, while the bottom row shows the zoomed region for better observation. From Figure 3, we can see that FreCaS effectively synthesizes the described contents while maintaining a coherent scene structure. DirectInference, MultiDiffusion and AttnEntropy often produce duplicated objects, such as the many astronauts and warriors. ScaleCrafter and HiDiffusion achieve reasonable image contents in experiments of SD2.1 but generate unnatural layouts in the experiments of SDXL, such as the excessive flowers on the ceiling in $\times 4$ experiment. Our FreCaS consistently maintains coherent image contents and layout in experiments of both SD2.1 and SDXL. AccDiffusion and DemoFusion also achieve natural image contents, but FreCaS generates clearer details such as the flowers and trains. Please refer to Appendix E for more visual results, including images with other aspect ratios.

### 4.3 EXPERIMENTS ON SD3

SD3 (Esser et al., 2024) adopts a rather different network architecture from SD2.1 and SDXL, and many existing methods cannot be applied. We can only compare FreCaS with DirectInference and DemoDiffusion Du et al. (2024). Due to page limitation, please refer to Appendix F for the results.

### 4.4 ABLATION STUDIES

In this section, we conduct ablation studies on $\times 4$ experiments of SDXL to investigate the effectiveness and settings of our cascaded framework, FA-CFG and CA-maps strategies.

**Effectiveness of each component.** We conduct a series of ablation studies to better demonstrate the effectiveness of each component of FreCaS, including the cascaded sampling framework, FA-CFG and CA-maps reutilization strategies. Please refer to Appendix G for more details.

**FA-CFG strategy.** The FA-CFG strategy aims to guide the model to generate content within the expanded frequency band. To achieve this, FA-CFG introduces two parameters, $w_l$ and $w_h$, to adjust the guidance strength on the low and high frequency components, respectively. When $w_l = w_h$, the FA-CFG strategy degenerates to the conventional CFG approach. We conduct a series of

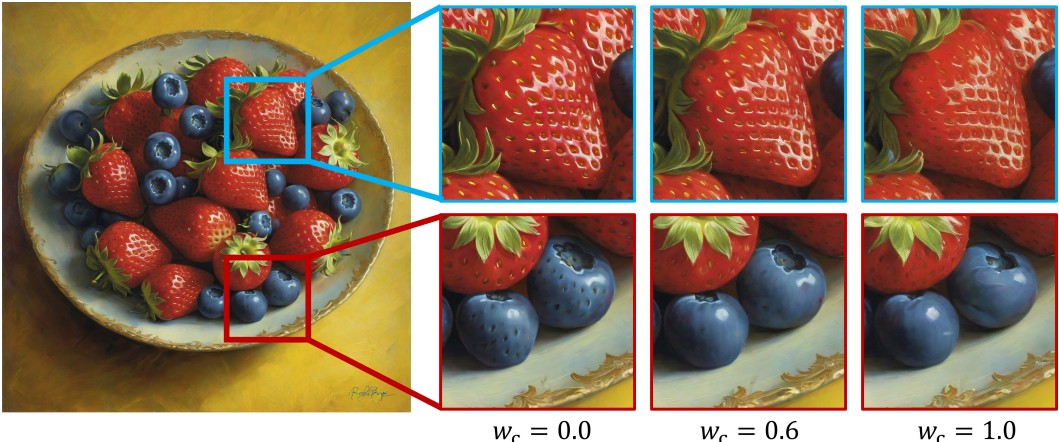

$$w_c = 0.0 \qquad w_c = 0.6 \qquad w_c = 1.0$$

Figure 6: Visual results on adjusting $w_c$ in CA-maps reutilization. The prompt is "Blueberries and Strawberries Art Print".

experiments to explore the optimal settings of the two parameters. First, we fix $w_l$ at 7.5 and vary $w_h$. The results are shown in Figure 4(a). We observe that as $w_h$ increases from 1.0 to 45, the $FID_b$ and $FID_p$ metrics initially decrease, indicating improved image quality. However, as $w_h$ becomes too high, the metrics begin to deteriorate. The sweet spot lies between 25 and 35, achieving a low $FID_b$ of nearly 12.65 and a low $FID_p$ of 17.91. We then fix $w_h$ at 35 and vary $w_l$. The results are presented in Figure 4(b). Reducing $w_l$ below 7.5 leads to a slight increase in $FID_p$ from 17.91 to 18.06, whereas increasing $w_l$ over 7.5 deteriorates $FID_r$ from 12.81 to 13.00. Compared to $w_h$, adjusting $w_l$ brings much smaller effects on those two metrics. Thus, we set $w_l$ to 7.5 for experiments on SD2.1 and SDXL, and set it to 7.0 for SD3.

Figure 5 provides visual examples of adjusting $w_h$. Increasing $w_h$ enhances the sharpness of details, such as clearer hair strands and more vivid flower petals. However, an excessively high value of $w_h$ (*e.g.*, 45) will introduce artifacts, as highlighted by the red boxes in the figure. This underscores the importance of selecting an appropriate $w_h$ value to strike a balance between detail enhancement and artifact suppression. Based on these findings, we set $w_l$ to 7.5 and $w_h$ to 35 yields favorable outcomes in most of the cases.

**CA-maps re-utilization.** To evaluate the effect of weight $w_c$ in the re-utilization of CA-maps, we conduct an ablation study by varying $w_c$ from 0 to 1. The results are shown in Figure 4(c). Increasing $w_c$ continuously decreases $FID_b$ but increases $FID_p$, indicating an improvement on the image layout but a drop on image details. To balance between the two metrics, we set $w_c = 0.6$. A visual example is shown in Figure 6. We see that this setting leads to a clearer textures on strawberry compared to $w_c = 1.0$ and prevents the unreasonable surface of the blueberry in $w_c = 0.0$.

**Inference schedule.** FreCaS uses two factors to adjust the inference schedule. The first one is the count of additional stages $N$. The second factor is the timestep $L$ of last latent in each stage. We conduct experiments on the selection of these two factors. The details can be found in Appendix G. Based on results, we set $L$ to 200, and set $N$ to 2 for $\times 4$ experiments and 3 for $\times 16$ experiments.

## 5 CONCLUSION

We developed a highly efficient **Fre**quency-aware **Ca**scaded **S**ampling framework, namely **FreCaS**, for training-free higher-resolution image generation. FreCaS leveraged the coarse-to-fine nature of latent diffusion process, reducing unnecessary computations in processing premature high-frequency details. Specifically, we divided the entire sampling process into several stages having increasing resolutions and expanding frequency bands, progressively generating image contents of higher frequency details. We presented a **F**requency-**A**ware **C**lassifier-**F**ree **G**uidance (**FA-CFG**) strategy to enable diffusion models effectively adding details of the expanded frequencies, leading to clearer textures. In addition, we fused the cross-attention maps of previous stages and current one to maintain consistent image layouts across stages. FreCaS demonstrated advantages over previous methods in both image quality and efficiency. In particular, with SDXL, it can generate a high quality $4096 \times 4096$ resolution image in 86 seconds on an A100 GPU.

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
