# Appendix to "FreCaS: Efficient Higher-Resolution Image Generation via Frequency-aware Cascaded Sampling"

In this appendix, we provide the following materials:

A Details of timestep shifting in the transition process (referring to Sec. 3.2 in the main paper);
B The detailed settings of FreCaS on $\times 4$ and $\times 16$ generation for SD2.1, SDXL and SD3 (referring to Sec. 4.1 in the main paper);
C Results of user studies and non-reference image quality assessment (NR-IQA) (referring to Sec. 4.1 in the main paper);
D Comparison with training-based methods and super-resolution methods (referring to Sec. 4.2 in the main paper);
E More visual results and visual comparisons (referring to Sec. 4.2 in the main paper);
F Experimental results of generation of SD3 (referring to Sec. 4.3 in the main paper);
G Ablation studies on individual components of FreCaS and inference schedule (referring to Sec. 4.4 in the main paper).

## A  SHIFTING TIMESTEP IN THE TRANSITION PROCESS

As mentioned in Sec. 3.2 of the main paper, FreCaS employs a five-step transition process to transform the last latent in the current stage $z_L^{s_{i-1}}$ to the first latent in the next stage $z_F^{s_i}$. In addition to changing the resolution, we adjust the timestep from $L$ to $F$ to ensure that the signal-to-noise ratio (SNR) (Kingma et al., 2021) could be a constant in the transition process. Given a state $z_t = \sqrt{\alpha_t} z_0 + \sqrt{1 - \alpha_t} \epsilon$ at timestep $t$, the SNR is defined as $\text{SNR}(z_t) = \frac{\alpha_t}{1 - \alpha_t}$, where $\alpha_1, \ldots, \alpha_T$ represent the noise schedule, and $\epsilon$ is Gaussian noise. It has been found (Hoogeboom et al., 2023; Chen, 2023) that the SNR maintains a consistent ratio across resolutions for diffusion models using the same noise schedule:

$$\text{SNR}(z_t^s) = \text{SNR}(z_t^{\hat{s}}) \cdot \left(\frac{s}{\hat{s}}\right)^\gamma,$$

where $s$ and $\hat{s}$ denote different resolutions. The value of $\gamma$ is typically set to 2.

Teng et al. (2024) and Gu et al. (2023) proposed to redesign the noise schedule to keep SNR consistent when changing the resolutions of intermediate states. Since the pre-trained diffusion models have fixed noise schedules, in this paper we adjust the timestep, instead of the noise schedule, to ensure consistent SNR between $z_L^{s_{i-1}}$ and $z_F^{s_i}$:

$$\text{SNR}(z_L^{s_{i-1}}) = \text{SNR}(z_F^{s_i}) \Rightarrow F = \alpha^{-1} \left( \frac{\left(\frac{s_{i-1}}{s_i}\right)^\gamma \cdot \alpha_L}{1 + \left(\left(\frac{s_{i-1}}{s_i}\right)^\gamma - 1\right) \cdot \alpha_L} \right), \tag{1}$$

where $\alpha^{-1}$ is the inverse function of $\alpha_t$. Proper adjustment of $\gamma$ can yield additional improvements.

Besides, SD3 (Esser et al., 2024) employs a similar formula to shift the timestep when varying resolutions:

$$F = \frac{\sqrt{\frac{s_i}{s_{i-1}}} \cdot L}{1 + (\sqrt{\frac{s_i}{s_{i-1}}} - 1) \cdot L}. \tag{2}$$

## B  EXPERIMENTAL SETTING DETAILS

The experimental setting details of our FreCaS are listed in Table 1.

## C  RESULTS OF USER STUDIES AND NR-IQA METRICS

We have (a) conducted user studies and (b) employed non-reference image quality assessment (NR-IQA) metrics to further assess the performance of FreCaS and its competing methods.

Table 1: Detailed settings of FreCaS on the experiments. $N$ denotes the count of additional stages. "Steps" presents the sampling steps in each stage. $L$ presents the timestep of last latent in each stage except for the final one. $\gamma$ denotes the SNR ratio in the transition process. $w_l$, $w_h$ and $w_c$ are the hyper-parameters of the proposed FA-CFG and CA-maps re-utilization.

| ﹀ | ﹀ | $N+1$ | Steps | $L$ | $\gamma$ | $w_l$ | $w_h$ | $w_c$ |
|---|---|---|---|---|---|---|---|---|
| SD2.1 | ×4 | 2 | 40,10 | 100 | 3.0 | 7.5 | 45.0 | 0.6 |
| | ×16 | 3 | 30,10,10 | 200,200 | 3.0 | 7.5 | 35.0 | 0.4 |
| SDXL | ×4 | 2 | 40,10 | 200 | 1.5 | 7.5 | 35.0 | 0.6 |
| | ×16 | 3 | 30,5,15 | 400,200 | 2.0 | 7.5 | 35.0 | 0.6 |
| SD3 | ×4 | 2 | 20,8 | 50 | - | 7.0 | 35.0 | 0.5 |

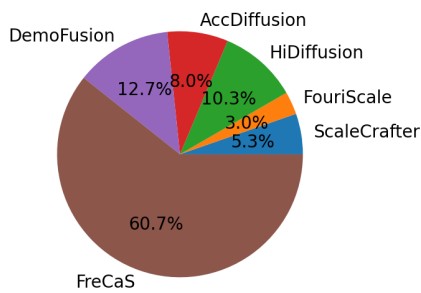

Figure 1: User study results on ×4 generation of SDXL.

Table 2: NR-IQA metrics on ×4 and ×16 generation of SDXL.

| Methods | ×4 | | | ×16 | | |
|---|---|---|---|---|---|---|
| | clipiqa↑ | niqe↓ | musiq↑ | clipiqa↑ | niqe↓ | musiq↑ |
| DirectInference | 0.522 | 4.167 | 53.98 | 0.469 | 4.370 | 29.00 |
| AttnEntropy | 0.547 | 4.210 | 54.87 | 0.528 | 4.614 | 27.98 |
| ScaleCrafter | 0.664 | 3.577 | 61.12 | 0.618 | 3.783 | 36.00 |
| FouriScale | 0.662 | 3.580 | 60.77 | 0.612 | 3.791 | 35.52 |
| HiDiffusion | **0.690** | 4.049 | 61.69 | 0.574 | 7.348 | 36.71 |
| AccDiffusion | 0.627 | 3.641 | 57.02 | 0.626 | 3.587 | 31.83 |
| DemoFusion | 0.651 | 3.410 | 58.98 | 0.637 | 3.376 | 33.46 |
| **Ours** | 0.668 | **3.391** | **63.10** | **0.646** | **3.367** | **37.33** |

## C.1   USER STUDIES

For the user studies, we compare FreCaS with ScaleCrafter, FouriScale, HiDiffusion, DemoFusion, and AccDiffusion on 2048×2048 image generation using SDXL. We randomly selected 30 prompts and generated one image per method for each prompt, creating 30 sets of images. Ten volunteers participated in the test, and they were asked to select the image with the best details and reasonable semantic layout from each set. The results are shown in Figure 1. We can see that FreCaS significantly outperforms other methods, with 60% votes as the best method. DemoFusion, AccDiffusion, and HiDiffusion perform similarly, with each having about 10% of the votes. In contrast, FouriScale and ScaleCrafter have the fewest votes, about 5% each.

## C.2   NR-IQA METRICS

For the NR-IQA metrics, we employ CLIPIQA (Wang et al., 2023), NIQE (Mittal et al., 2012), and MUSIQ (Ke et al., 2021) on ×4 and ×16 image generations with SDXL. The results are presented in

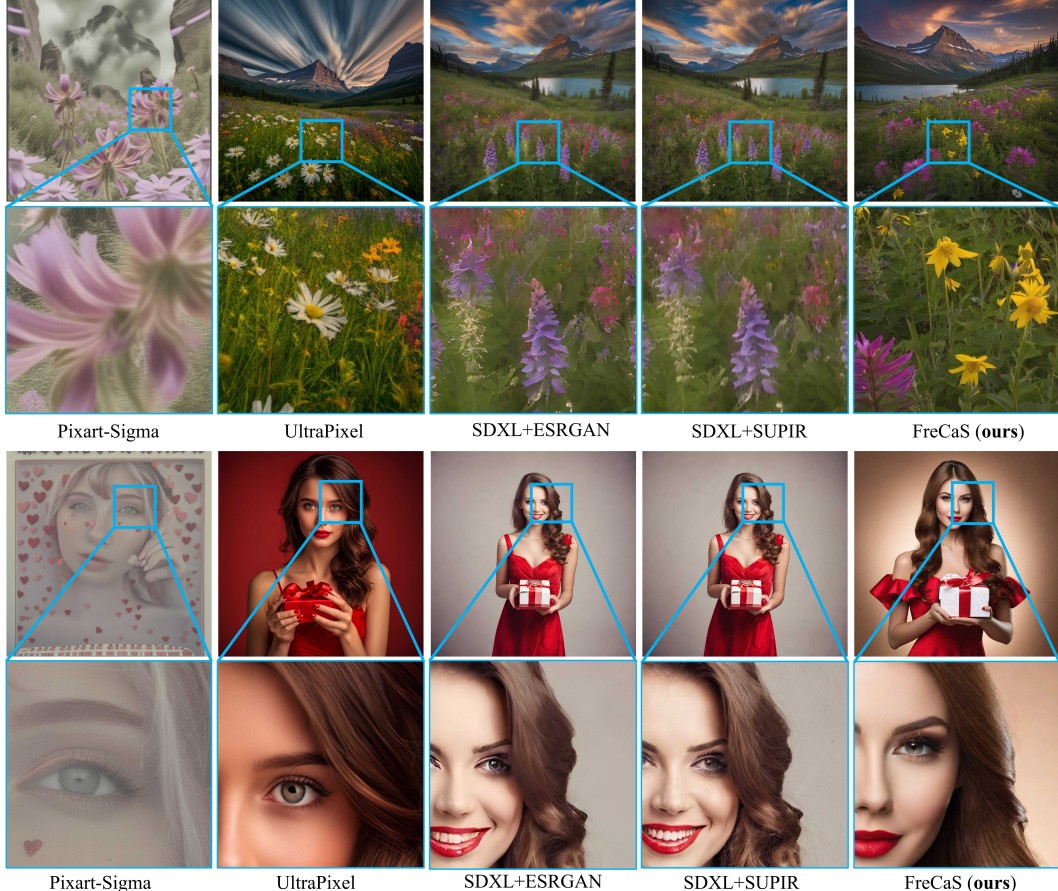

| Pixart-Sigma | UltraPixel | SDXL+ESRGAN | SDXL+SUPIR | FreCaS (**ours**) |

| Pixart-Sigma | UltraPixel | SDXL+ESRGAN | SDXL+SUPIR | FreCaS (**ours**) |

Figure 2: Visual comparison with training-based methods and super-resolution methods on ×4 generation of SDXL.

Table 2. Our FreCaS consistently outperforms all the other methods. For example, on ×4 generation, FreCaS achieves a CLIPIQA score of 0.668, a NIQE score of 3.391, and a MUSIQ score of 63.10, compared to 0.651, 3.410, and 58.98 for DemoFusion. On ×16 generation, FreCaS achieved a CLIPIQA score of 0.646, a NIQE score of 3.367, and a MUSIQ score of 37.33, compared to 0.626, 3.587, and 31.83 for AccuDiffusion. Notably, FreCaS only lags behind HiDiffusion on the CLIPIQA metric in ×4 image generation.

# D COMPARISON WITH TRAINING-BASED METHODS AND SUPER-RESOLUTION METHODS

## D.1 QUANTITATIVE AND VISUAL COMPARISON

We conducted additional experiments comparing FreCaS with training-based methods (Pixart-Sigma (Chen et al., 2024) and UltraPixel (Ren et al., 2024)) and super-resolution methods (ESRGAN (Wang et al., 2021) and SUPIR (Yu et al., 2024)). To ensure fair comparisons, we set the model precision to fp16 (bf16 for UltraPixel, as recommended by the authors) and use the DDIM sampler for diffusion-based methods. For Pixart-Sigma, we can only report its results for 2048×2048 image generation since its 4K model is not available. The quantitative results are summarized in Table 3.

From Table 3, we can see that FreCaS outperforms Pixart-Sigma and UltraPixel in most metrics. For example, FreCaS achieves an FID score of 16.48 and an IS score of 17.18, compared to 26.1 and 14.44 of Pixart-Sigma, and 25.56 and 17.11 of UltraPixel on the ×4 image generation task. This is because Pixart-Sigma, as acknowledged by the authors, heavily relies on the advanced samplers

Table 3: Comparison with training-based methods and super-resolution methods on $\times 4$ and $\times 16$ generation of SDXL.

| | Methods | FID$\downarrow$ | FID$_p\downarrow$ | IS$\uparrow$ | IS$_p\uparrow$ | CLIP SCORE$\uparrow$ | Latency(s)$\downarrow$ |
|---|---|---|---|---|---|---|---|
| $\times 4$ | Pixart-Sigma | 26.11 | 38.58 | 14.44 | 14.45 | 28.10 | 71.45 |
| | UltraPixel | 25.56 | 19.95 | 17.11 | 17.10 | 33.17 | 41.70 |
| | SDXL+ESRGAN | 13.03 | 18.10 | 17.30 | 16.58 | 34.13 | 6.36 |
| | SDXL+SUPIR | 12.08 | 17.31 | 17.57 | 17.12 | 34.16 | 105.5 |
| | **Ours** | 16.48 | 17.91 | 17.18 | 17.31 | 33.28 | 13.84 |
| $\times 16$ | UltraPixel | 51.43 | 45.88 | 12.48 | 13.73 | 33.07 | 162.4 |
| | SDXL+ESRGAN | 45.86 | 43.10 | 12.94 | 13.48 | 33.44 | 7.25 |
| | SDXL+SUPIR | 43.94 | 39.35 | 13.22 | 14.37 | 33.49 | 512.4 |
| | **Ours** | 42.75 | 39.82 | 12.68 | 14.16 | 33.03 | 85.87 |

Table 4: Stability experiments on 200 images of 20 prompts on $\times 4$ generation.

| Methods | clipiqa$\uparrow$ | | | niqe$\downarrow$ | | | musiq$\uparrow$ | | |
|---|---|---|---|---|---|---|---|---|---|
| | Mean | AoS | SoM | Mean | AoS | SoM | Mean | AoS | SoM |
| Pixart-Sigma | 0.558 | 0.05 | 0.11 | 5.256 | 0.35 | 0.95 | 51.546 | 4.49 | 8.24 |
| UltraPixel | 0.540 | 0.04 | 0.11 | 4.625 | 0.42 | 1.55 | 56.215 | 2.94 | 7.95 |
| FreCaS | 0.633 | 0.11 | 0.04 | 3.886 | 0.87 | 0.27 | 59.756 | 9.64 | 2.95 |

(see https://github.com/PixArt-alpha/PixArt-sigma/issues/65) so that the results are not very stable. UltraPixel, while achieving comparable performance to DemoFusion, still lags behind FreCaS in most metrics. Besides, the two methods are much slower than our FreCaS.

For SR-based methods, FreCaS may have lower FID, IS, and CLIP scores than SDXL+ESRGAN. This is because SR methods are designed to strictly adhere to low-resolution inputs, while these metrics (FID, IS, and CLIP) evaluate images by downsampling them to low resolution, which cannot well reflect the quality of generated high-resolution images. However, FreCaS significantly outperforms SDXL+ESRGAN in FID$_p$ and IS$_p$. Specifically, FreCaS achieves an FID$_p$ score of 39.82 and an IS$_p$ score of 14.16, compared to 43.10 and 13.48 of SDXL+ESRGAN on $\times 16$ image generation. This indicates its superior ability to generate high-resolution local details. This observation is consistent with the findings in the DemoFusion paper. Additionally, SDXL+SUPIR outperforms FreCaS in FID$_p$ and IS$_p$, but at the cost of much longer inference latency (85.87 seconds for FreCaS vs. 512.4 seconds for SDXL+SUPIR on $\times 16$ image generations).

We have provided some visual comparisons in Figure 2. One can see that FreCaS demonstrates better visual quality than either training-based or SR-based methods in high-resolution image generation, such as the more vivid and clearer flowers, hairs and the more natural color of lips.

## D.2 STABILITY METRICS

To quantitatively analyze the stability of training-based and training-free methods, we generated 200 images for 20 randomly selected prompts (10 images for each prompt) using Pixart-Sigma (with default sampler unless otherwise stated), UltraPixel, and our FreCaS. Considering that FID and IS are not suitable for evaluating individual examples, we adopt the NR-IQA metrics (CLIPIQA, NIQE, and MUSIQ) to measure the performance of each method. In specific, we define the following three measures to evaluate the generation quality, stability and consistency of each method.

- **Average Score (Mean):** The average score across the 200 generated images for each of the three metrics (CLIPIQA, NIQE, and MUSIQ). This metric can reflect the generation quality of each method.

Table 5: **User studies of visual quality and success rate on $\times 4$ generation with SDXL.**

| Methods | Image Quality | | Success Rate | |
|---|---|---|---|---|
| | Counts | Percentage | Counts | Percentage |
| Pixart-Sigma | 5 | 5% | 52 | 20.8% |
| UltraPixel | 37 | 37% | 96 | 38.4% |
| **Ours** | 58 | 58% | 68 | 27.2% |

- **Average of Standard Deviations (AoS):** We first compute the standard deviation of the metrics for each prompt across 10 runs, and then report the average of these standard deviations across all 20 prompts. This metric can reflect the stability of each method.

- **Standard Deviation of Averages (SoM):** We first compute the mean of the metrics for each prompt across 10 runs, and then report the standard deviation of these mean values across all 20 prompts. This metric can reflect the consistency of a method's performance across different prompts.

The results are listed in Table 4. From this table, we can see that our FreCaS achieves the highest "Mean" scores across the three metrics, demonstrating the best performance in term of generation quality. Pixart-Sigma and UltraPixel have smaller AoS scores than FreCaS, indicating better stability for the same input prompt. However, FreCaS demonstrates significantly better SoM scores than Pixart-Sigma and UltraPixel, indicating that it can consistently achieve better results across various prompts.

### D.3    USER STUDIES ON VISUAL QUALITY AND SUCCESS RATE

We conducted user studies to explore the generated image quality and success rate of Pixart-Sigma, UltraPixel, and our FreCaS. The results are listed in Table 5.

For the study on generation quality, we randomly select 20 prompts from Laion5B and generate one image per method for each prompt, creating 20 sets of images (3 images per set). Five volunteers (3 males and 2 females) were invited to participate in the test. All the volunteers are not working in the area of image generation to avoid potential bias. Each time, the set of 3 images for the same prompt are presented to the volunteers in random order. The volunteers can view the images multiple times, and they are asked to select the image with the best quality from each set. There are 100 votes in total.

For the study on success rate, we randomly select 10 prompts and generate five images per prompt for each method, resulting in 50 images per method. We invited the same five volunteers as in the study of generation quality to judge whether the generated image is a success or failure. When making the decision, the volunteers are instructed to consider two factors. First, whether the image content is faithful to the description of the prompt. Second, whether the image quality is satisfactory. Only when both the two requirements are met, the generation is considered as a success. There are 250 judges for each method.

As we can see from Table 5, our FreCaS outperforms significantly Pixart-Sigma and UltraPixel in terms of image generation quality, with 58% of the images being voted as the best. In terms of success rate, UltraPixel works the best, with 96 out of 250 images being marked as successful. Our FreCaS lags behind, with 68 successful cases. However, our FreCaS still generates more successful results than Pixart-Sigma (52 images), indicating that a well designed training-free method can surpass some training-based methods. Furthermore, we can also observe that none of the methods, including training-based and training-free ones, achieves a success rate higher than 40%. This implies that there are much space to improve.

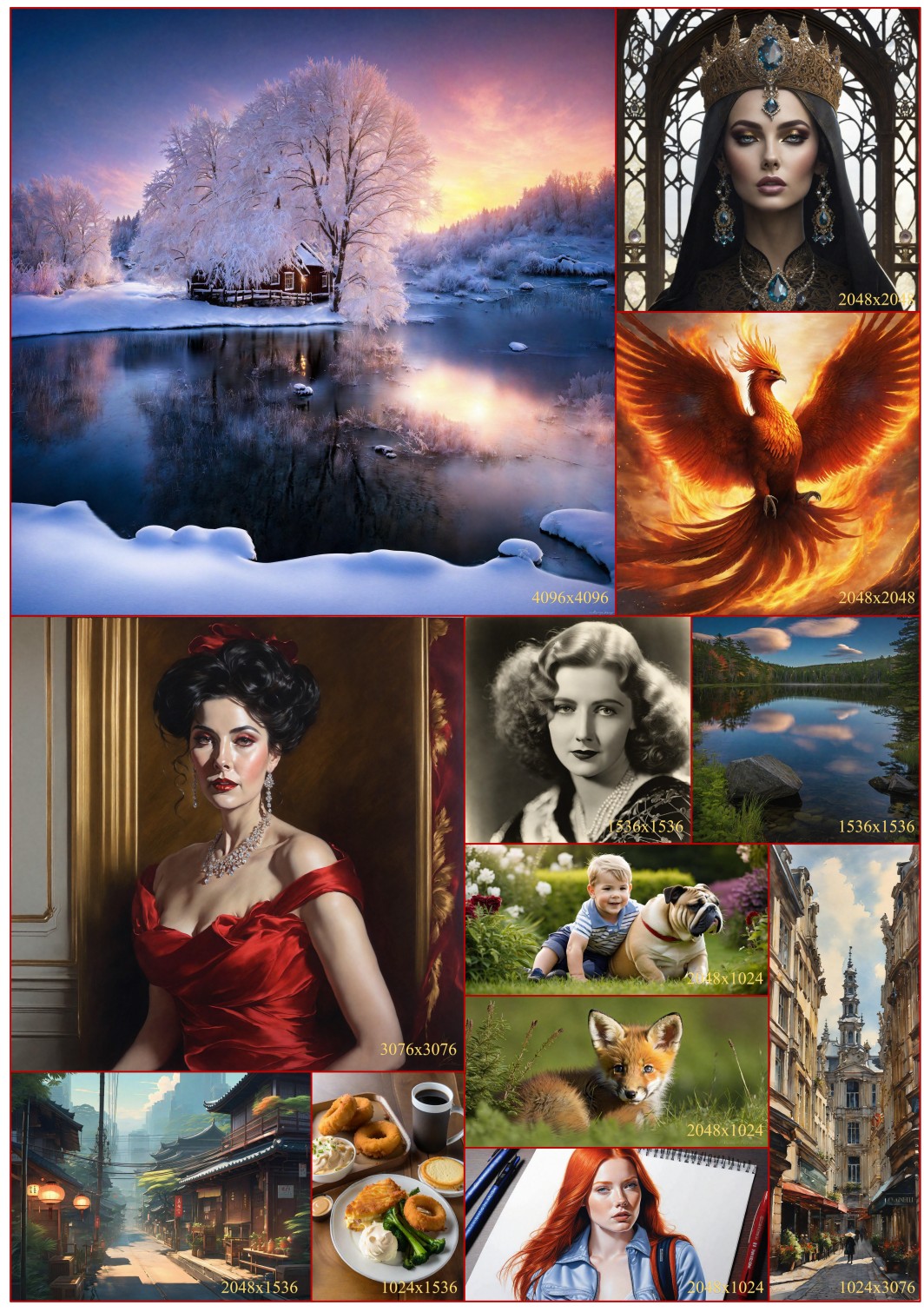

Figure 3: Visual results of FreCaS on SDXL. Please zoom-in for better view.

Table 6: Experiments on ×4 generation of SD3.

| Methods | $FID_b\downarrow$ | $FID_p\downarrow$ | IS↑ | $IS_p\uparrow$ | CLIP SCORE↑ | Latency (s)↓ | SpeedUP↑ |
|---------|-------|-------|------|------|------------|-------------|----------|
| DirectInference | 35.68 | 45.35 | 12.52 | 12.60 | 31.45 | 38.53 | 1× |
| Demodiffusion | 15.19 | 44.34 | 17.84 | 14.99 | 31.09 | 63.33 | 0.61× |
| **Ours** | 9.76 | 26.62 | 17.83 | 16.72 | 31.17 | 15.94 | 2.42× |

# E   MORE VISUAL RESULTS

## E.1   MORE VISUAL RESULTS

Figure 3 illustrates more visual results of FreCaS, including those with varying aspect ratios. From top to bottom, and left to right, the prompts used in examples are: 1. "Beautiful winter wallpapers." 2. "A regal queen adorned with jewels." 3. "A majestic phoenix, wings ablaze, rising from ashes, the flames casting a warm glow." 4. "Lady in Red oil portrait painting won the John Singer Sargent People's award." 5. "Star of the day – Actress Evelyn Laye - 1917." 6. "Photograph - Clouds Over Daicey Pond by Rick Berk." 7. "little-boy-with-large-bulldog-in-a-garden-france." 8. "03-Brussels-Maja-Wronska-Travels-Architecture-Paintings.", 9. "Red Fox Pup Print by William H. Mullins." 10. "Lovely Illustrations Of Cityscapes Inspired By Southeast Asia Malaysian digital illustrator Chong Fei Giap's illustrations of cityscapes are lovely and inspiring. Fantasy Landscape, Landscape Art, Illustrator, Japon Tokyo, Animation Background, Art Background, Matte Painting, Anime Scenery, Jolie Photo." 11. "A plate with creamy chicken and vegetables, a side of onion rings, a cup of coffee and a slice of cheesecake." 12. "Hyper-Realistic Portrait of Redhead Girl Drawn with Bic Pens."

To further validate the performance of FreCaS in real-world application scenarios, we have provided additional visual results in three categories:

- **Simple scenes.** These images typically contain a single object in a realistic style. We display images of people, animals, landscapes, buildings, and other common objects. The visual results for this group are presented in Figure 4.
- **Various styles.** This group showcases images in different artistic styles, including oil painting, pencil sketch, ink wash, watercolor, and poster art. The results are shown in the first two rows of Figure 5.
- **Complex scenes.** These images contain multiple objects or have intricate textures. The results are displayed in the bottom two rows of are presented in Figure 5.

From these visual results, it is evident that FreCaS consistently generates high-quality images across various styles and contents, demonstrating the capability of FreCaS in real-world applications.

## E.2   MORE VISUAL COMPARISONS

We show more visual comparisons in Figure 6. From top to bottom, the prompts used in the four groups of examples are: 1. "A small den with a couch near the window." 2. "A painting of a candlestick holder with a candle, several pieces of fruit and a vase, with a gold frame around the painting." 3. "A noble knight, riding a white horse, the castle gates opening." 4. "Mystical Landscape Digital Art - Lonely Tree Idyllic Winterlandscape by Melanie Viola."

We have provided more 4K visual comparisons under realistic scenes in Figure 7. As can be seen, our FreCaS consistently delivers better results in both image layout and semantic details.

# F   EXPERIMENTS ON SD3

In this section, we present the results of the ×4 generation experiments on SD3. SD3 employs a transformer-based denoising network. It eliminates all convolutional layers, thereby preventing the application of many existing methods, such as ScaleCrafter and FouriScale. Besides, SD3 exhibits fine details in the central region but shows corrupted textures in the surrounding regions (see

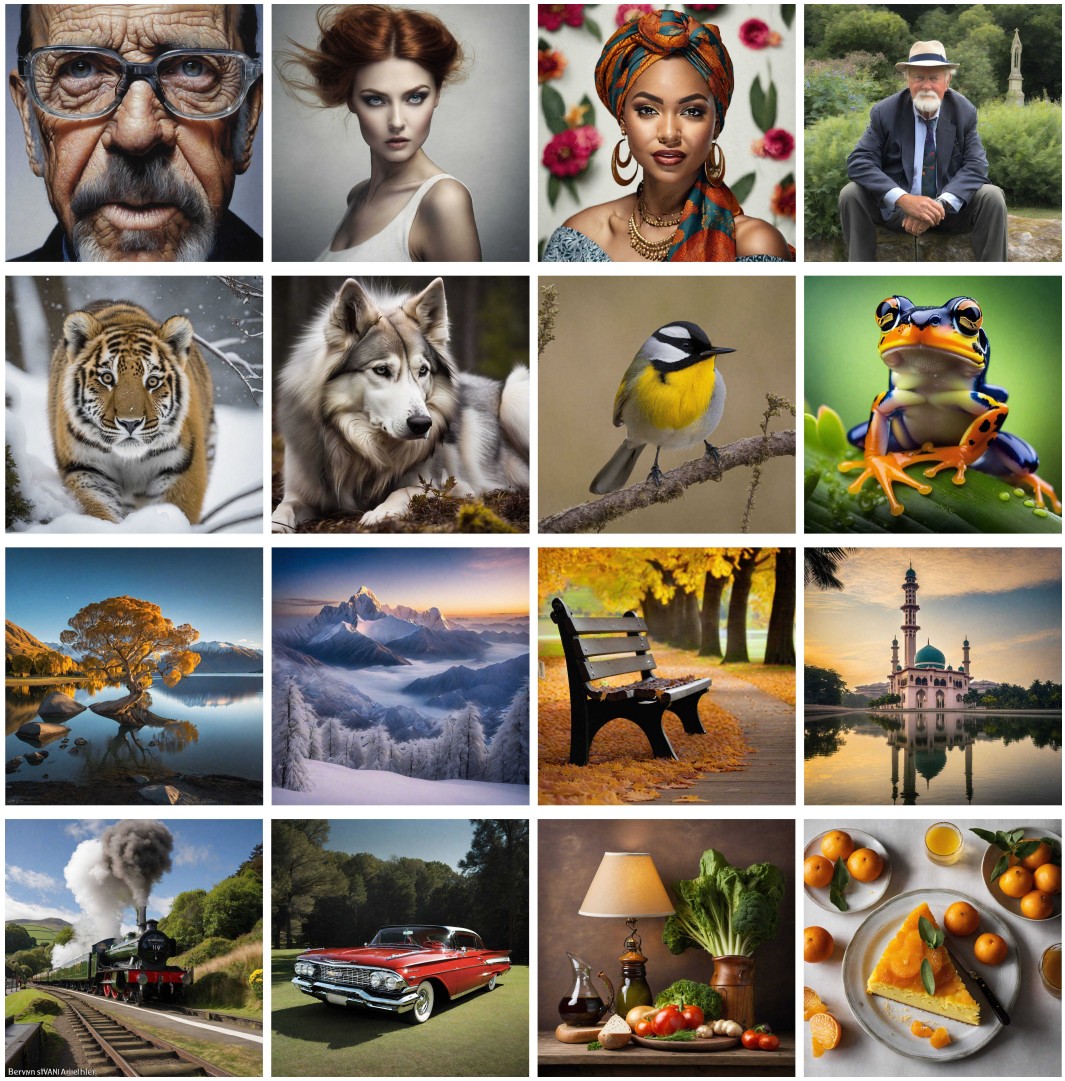

Figure 4: More visual results on simple scenes.

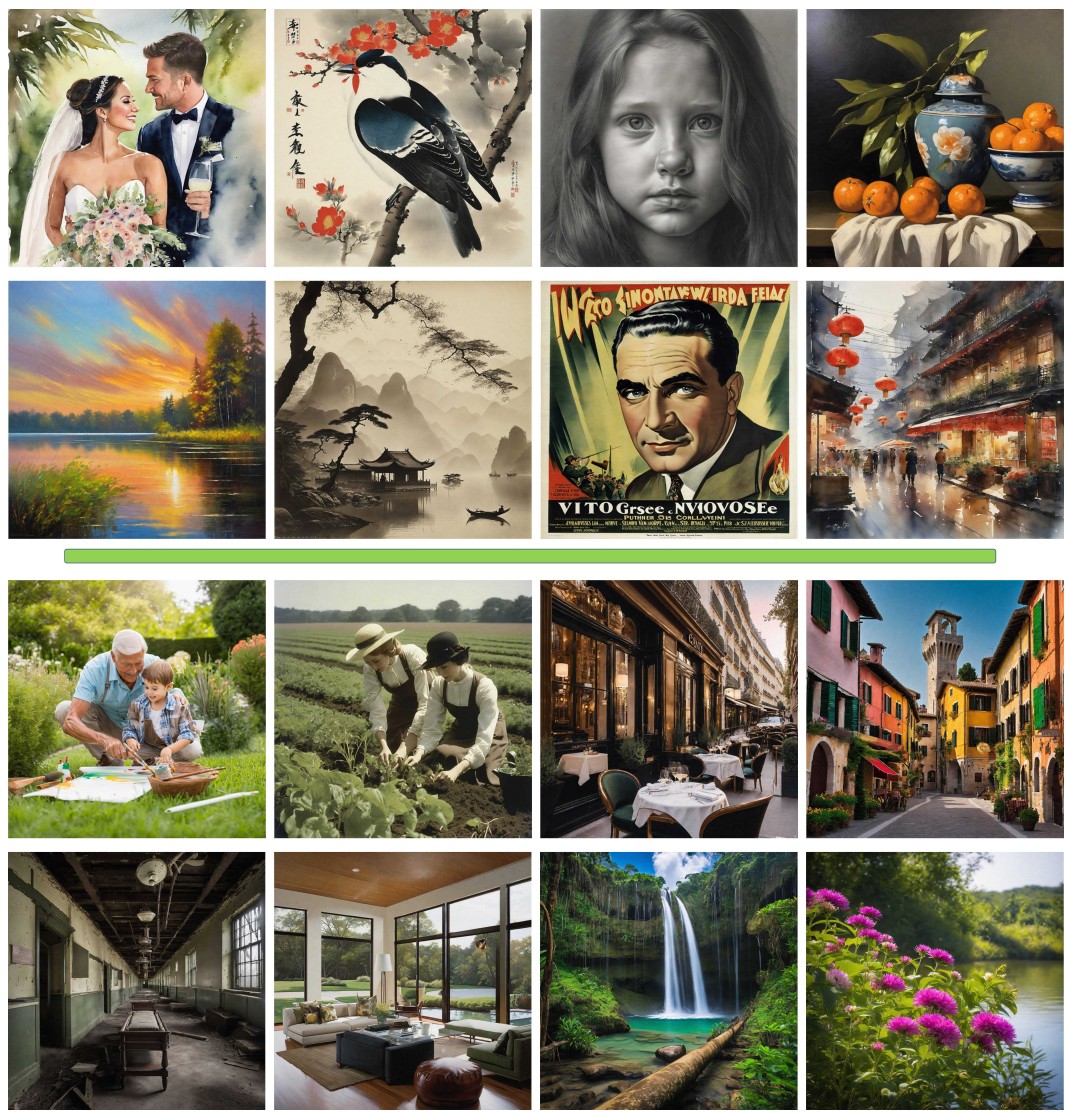

Figure 5: More visual results of various styles (top two rows) and complex scenes (bottom two rows).

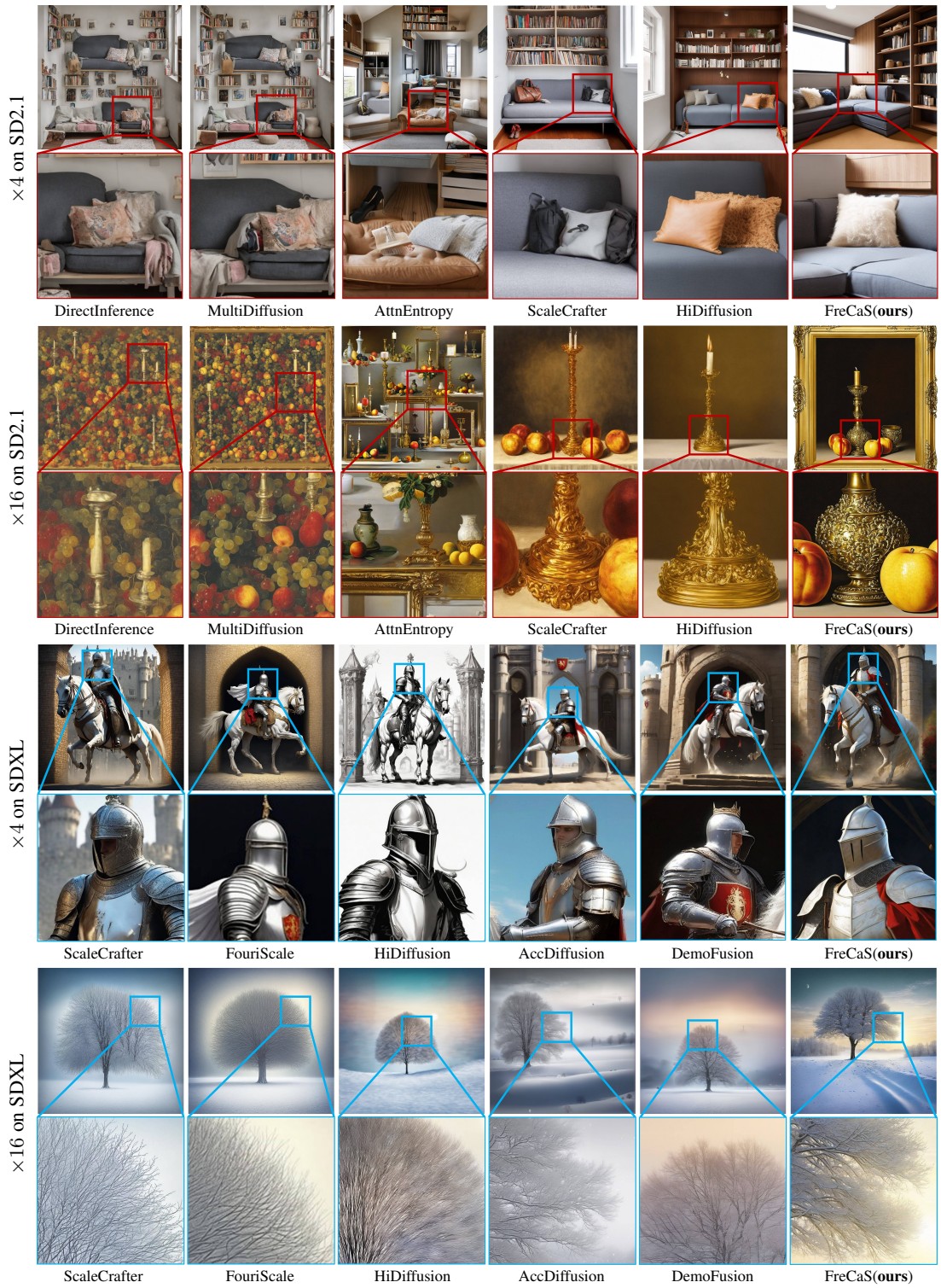

Figure 6: Visual comparisons on ×4 and ×16 experiments of SD2.1 and SDXL. Please zoom-in for better view.

Figure 8). This issue with the image layout also significantly impacts the performance of other methods, such as DemoFusion. Therefore, we only compare our FreCaS with DirectInference and DemoFusion. Table 6 and Figure 8 present the quantitative and qualitative results, respectively.

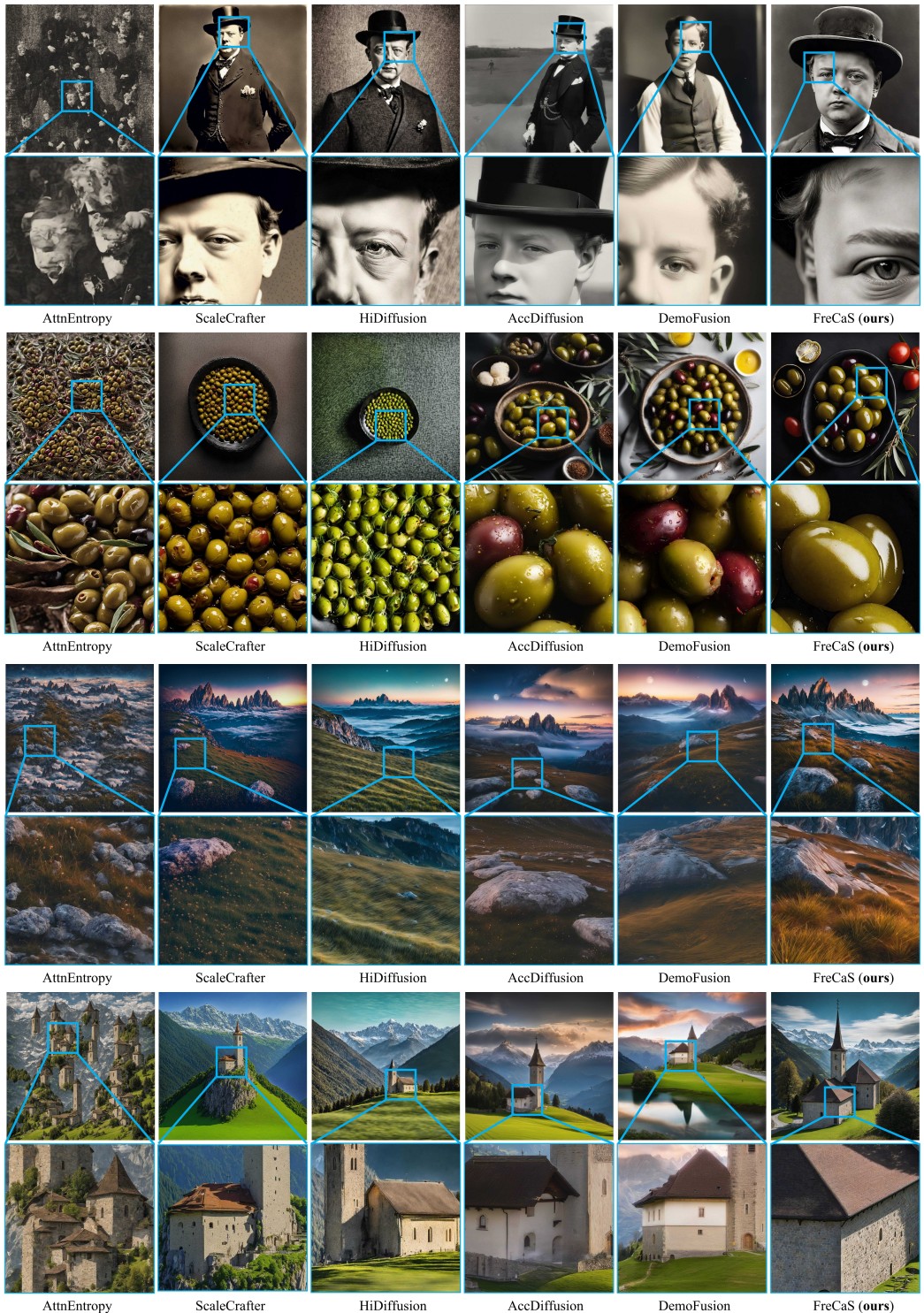

Figure 7: More 4K comparisons in realistic styles. From top to bottom, the prompts are "Young winston churchill.", "Olive food photography.", "Mountains in fog at beautiful night. Dreamy landscape with mountain peaks, stones, grass, blue sky with blurred low clouds, stars and moon. Rocks at dusk." and "Image Church Switzerland towers San Romerio Nature Mountains Scenery Made of stone Tower mountain landscape photography."

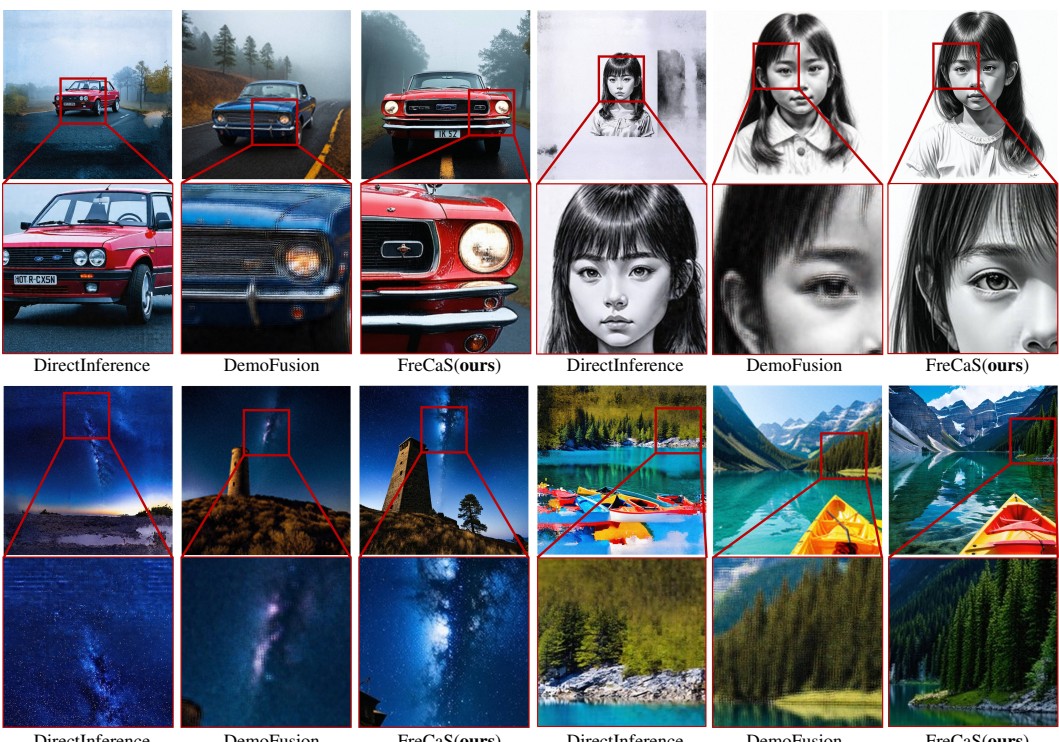

Figure 8: Visual comparison on ×4 experiments of SD3. From top to bottom, from left to right, the prompts used in the four groups of examples are: 1. "Car Photograph - Ford In The Fog by Debra and Dave Vanderlaan." 2. "Rupert Young is Sir Leon in Merlin season 5 copy." 3. "Watchtower, Shooting Star & Milky Way, Gualala, CA." 4. "Colorful Autumn in Mount Fuji, Japan - Lake Kawaguchiko is one of the best places in Japan to enjoy Mount Fuji scenery of maple leaves changing color giving image of those leaves framing Mount Fuji.". Zoom-in for better view.

Table 7: Ablation studies on $2048 \times 2048$ generation of SDXL.

| Model | cascaded framework | FA-CFG | CA-reuse | FID↓ | $\text{FID}_p\downarrow$ | IS↑ | $\text{IS}_p\uparrow$ | CLIP SCORE↑ | Latency (s) |
|---|---|---|---|---|---|---|---|---|---|
| #1 | | | | 39.14 | 29.71 | 11.52 | 14.60 | 32.51 | 34.10 |
| #2 | ✓ | | | 17.62 | 20.49 | 17.01 | 16.54 | 33.24 | 13.71 |
| #3 | ✓ | ✓ | | 16.62 | 17.91 | 17.16 | 16.82 | 33.34 | 13.74 |
| #4 | ✓ | ✓ | ✓ | 16.48 | 17.91 | 17.18 | 17.31 | 33.28 | 13.84 |

From Table 6, it is evident that FreCaS achieves superior performance in terms of image quality and inference speed. Specifically, FreCaS achieves the best results on $\text{FID}_b$, $\text{FID}_p$, IS, and $\text{IS}_p$, and only slightly lags behind DirectInference in terms of CLIP score. Moreover, FreCaS generates a $2048 \times 2048$ image in about 16 seconds, achieving a speed-up of $2.42\times$ and $3.97\times$ compared to DirectInference and DemoFusion, respectively. Figure 8 illustrates the generated images. Directly employing the pre-trained SD3 model to generate higher-resolution images, DirectInference leads to unreasonable image layout with the surrounding parts being corrupted, such as the road and trees. The results of DemoFusion exhibits strange artifacts, such as the car faces and eyes. In contrast, our FreCaS successfully maintains the natural image structure while obtaining fine details.

## G   ABLATION STUDIES ON INDIVIDUAL COMPONENTS AND INFERENCE SCHEDULE

We further conduct ablation studies to verify the effectiveness of each components and the settings of inference schedule of our FreCaS.

Table 8: Ablation studies on $N$ in FreCaS.

| $N$ | resolutions | $\text{FID}_b\downarrow$ | $\text{FID}_p\downarrow$ |
|---|---|---|---|
| 0 | 2048 | 43.83 | 29.71 |
| 1 | $1024 \rightarrow 2048$ | 12.63 | 17.91 |
| 2 | $1024 \rightarrow 1536 \rightarrow 2048$ | 41.36 | 28.68 |

Table 9: Ablation studies on $L$ in FreCaS.

| $L$ | $\text{FID}_b\downarrow$ | $\text{FID}_p\downarrow$ |
|---|---|---|
| 0 | 12.57 | 18.20 |
| 100 | 12.69 | 18.10 |
| 200 | 12.63 | 17.91 |
| 300 | 13.30 | 18.57 |
| 400 | 13.34 | 18.62 |

### G.1 EFFECTIVENESS OF EACH COMPONENT

To better verify the effectiveness of each component of FreCaS, we conducted more ablation studies on our proposed cascaded framework, FA-CFG, and CA-reuse strategies. The results are shown in Table 7. One can see that our cascaded framework significantly outperforms the baseline, with a decrease of 22.52 in the FID score and a reduction of 20.39 seconds in latency. This demonstrates the high efficiency of our proposed cascaded framework. Our FA-CFG strategy improves both FID and IS scores and shows substantial improvement in $\text{FID}_p$, demonstrating its effectiveness in generating realistic image details. The CA-reuse strategy further enhances $\text{IS}_p$, indicating its effectiveness in improving semantic appearance. Moreover, these strategies introduce minimal additional latency.

### G.2 EXPERIMENTS ON INFERENCE SCHEDULE

In this section, we conduct experiments on the selection of $N$ (number of additional stages) and $L$ (the timestep of last latent in each stage). The two factors are employed to adjust the inference schedule of our FreCaS. We reports the scores of $\text{FID}_b$ and $\text{FID}_p$ by varying the two factors in Table 8 and Table 9, respectively.

**Choice of $N$.** From Table 8, we see that $N = 1$ achieves an $\text{FID}_b$ score of 12.63 and an $\text{FID}_p$ score of 17.91, significantly better than $N = 0$ and $N = 2$ in the $\times 4$ generation task for SDXL. This could be attributed to the fact that a larger value of $N$ introduces more transition steps, which can lead to much information loss. Conversely, a smaller value of $N$ reduces the effectiveness of FreCaS, degenerating it to the DirectInference method.

**Choice of $L$.** From Table 9, we can see that a smaller $L$ improves $\text{FID}_b$ score but deteriorates $\text{FID}_p$. This is because the details generated at lower resolutions conflict with those at higher resolutions. Thus, we set $L$ to 200 to avoid generating excessive unwanted details in the early stages.