# OpenReview forum: "FreCaS: Efficient Higher-Resolution Image Generation via Frequency-aware Cascaded Sampling"
_ICLR.cc/2025/Conference — ICLR 2025 Poster_

### Official Review · Reviewer_Sncf · 2024-11-01

**Soundness:** 3
**Presentation:** 3
**Contribution:** 3
**Rating:** 6
**Confidence:** 4

**Summary:**

The authors introduce an efficient Frequency-aware Cascaded Sampling (FreCaS) framework for higher resolution image generation. FreCaS decomposes the sampling process into cascaded stages with gradually increased resolutions, progressively expanding frequency bands and refining the corresponding details. Additionally, the authors fuse the cross-attention maps of previous and current stages to
avoid synthesizing unfaithful layouts. Experiments demonstrate that FreCaS outperforms state-of-the-art methods in image quality and generation speed.

**Strengths:**

1. The  FreCaS is faster than ScaleCrafter and DemoFusion in generating a 2048×2048 image.
2. The  FreCaS introduces a FA-CFG strategy that assigns different guidance strengths to different frequency components.
3. The  FreCaS can be extended to more complex models, validating its broad applicability and versatility.

**Weaknesses:**

1.  FreCaS relies on several hyperparameters (e.g., w_l, w_h, w_c) that may need careful adjustment to achieve optimal results, which may increase the complexity of experiments.
2. Although FreCaS can be extended to more complex models, its performance may be limited by the pre-trained models.
3. This idea is somewhat similar to FreeU and FreeEnhance.

**Questions:**

1. In addition to metrics like FID and IS, more comprehensive image quality assessment indicators, such as human subjective evaluation, could be considered to assess the quality of generated images more thoroughly.
2. The experimental part may need to further explore the performance of FreCaS in real-world application scenarios, such as different image contents, styles, and complexities.
3. Although FreCaS improves efficiency, the demand for computational resources remains high when generating high-resolution images.

---

> ### Author Response · Authors · 2024-11-23
>
> We sincerely thank this reviewer for the recognition on the efficiency and generation of our FreCaS. Based on this reviewer's suggestion, we have conducted more experiments as well as a user study. Please find what below our itemized responses to this reviewer's questions.
>
> **Q1 Hyperparameter settings.**
>
> Thanks for the comment. Training-free based methods typically involve a few hyper-parameters to control the diffusion process. For example, DemoFusion relies on specific values of $\sigma_1$, $\sigma_2$, $\alpha_1$, $\alpha_2$, $\alpha_3$, and so on. To minimize the complexity, in FreCaS we provide two ways to set the hyperparameters:
>
> + **Default values for immediate use:** For all hyperparameters, we provide default values for specific applications. We generally recommend setting $ w_l $ to be consistent with the guidance scale, $ w_h $ to 35, and $ w_c $ to 0.6. These default values are effective across a wide range of scenarios.
> + **Ablation studies for parameter setting guidance:** In the paper, we conduct extensive ablation studies to explore the significance and impact of each hyperparameter on image quality. Specifically, larger $ w_h $ tends to generate higher-quality details but may introduce artifacts. $ w_c $ is primarily used to balance detail generation and semantic structure. Through these experiments, we provide clear guidance for adjusting these parameters based on specific requirements.
>
> While FreCaS does rely on hyperparameters, the provision of default values and the guidance for setting them ensure that the tuning process is manageable, which improves the practicality and usability of our method.
>
> **Q2 Performance limitation by pre-trained models.**
>
> Thanks for the comment. First of all, we agree with this reviewer that as a training-free method, FreCaS relies on the pre-trained models. A stronger pre-trained model will offer the adapted model a better foundation. However, we'd like to respectfully argue that the development of training-free methods is just to address the limitations of pre-trained models in various downstream applications. A well-designed training-free method, such as our FreCaS, can indeed improve much the performance of the pre-trained model. For example, both HiDiffusion and our FreCaS are effective in lifting SD2.1 and SDXL models in higher-resolution image generation.
>
> On the other hand, FreCaS demonstrates superior generation capabilities compared to existing training-free high-resolution image generation methods. In particular, FreCaS focuses on improving the general sampling process of diffusion models, so that it can be easily applied to more advanced yet more complex pretrained models like SD3, where DemoFusion and HiDiffusion fail to perform effectively.
>
> We hope our explanations can address this reviewer's concern.
>
> **Q3 Differences from FreeU and FreeEnhance.**
>
> Thanks for the comment. There are two main differences between FreCaS and FreeU and FreeEnhance.
>
> **First**, FreeU and FreeEnhance focus on enhancing the visual quality of generated images at base training sizes. In contrast, FreCaS focuses on improving the efficiency and effectiveness of generating higher-resolution images than the training sizes.
>
> **Second**, FreeU adjusts the weights of different frequency components of network features, and FreeEnhance adds lower-magnitude noise to the high-frequency component of latents. However, our FreCaS operates at the denoising scores by assigning higher CFG strength to higher frequency components.
>
> Actually, our FreCaS is orthogonal and complementary to those two methods. For example, integrating FreCaS with FreeU results in an improvement of 1.05 on the FID metrics without introducing additional latency on 2048$\times$2048 image generations using SDXL.

---

> > ### Author Response · Authors · 2024-11-23
> >
> > **Q4 More comprehensive image quality assessment.**
> >
> > Thanks for the nice suggestions. As suggested, we have (a) conducted a user study and (b) employed non-reference image quality assessment (NR-IQA) metrics to further assess the performance of FreCaS and its competing methods.
> >
> > **(a)** For the user studies, we compared FreCaS with ScaleCrafter, FouriScale, HiDiffusion, DemoFusion, and AccDiffusion on 2048×2048 image generations using SDXL. We randomly selected 30 prompts and generated one image per method for each prompt, creating 30 sets of images. Ten volunteers participated in the test, and they were asked to select the image with the best details and reasonable semantic layout from each set. The results are shown in Fig. 1 of the updated supplementary file. We can see that FreCaS significantly outperforms other methods, with 60\% votes as the best method. DemoFusion, AccDiffusion, and HiDiffusion perform similarly, with each having about 10\% of the votes. In contrast, FouriScale and ScaleCrafter have the fewest votes, about 5\% each.
> >
> > **(b)** For the NR-IQA metrics, we employ CLIPIQA, NIQE, and MUSIQ on $\times4$ and $\times 16$ image generations with SDXL. The results are presented in Table 1. Our FreCaS consistently outperforms all the other methods. For example, on $\times 4$ generation, FreCaS achieves a CLIPIQA score of 0.668, a NIQE score of 3.391, and a MUSIQ score of 63.10, compared to 0.651, 3.410, and 58.98 for DemoFusion. On $\times 16$ generation, FreCaS achieved a CLIPIQA score of 0.646, a NIQE score of 3.367, and a MUSIQ score of 37.33, compared to 0.626, 3.587, and 31.83 for AccuDiffusion. Notably, FreCaS only lags behind HiDiffusion on the CLIPIQA metric in $\times 4$ image generation.
> >
> > Tab. 1 Non-reference IQA metrics on $\times 4$ and $\times 16$ generation of SDXL.
> > | Methods            | ×4                  | ×4                  | ×4                  | ×16                 | ×16                 | ×16                 |
> > |--------------------|---------------------|---------------------|---------------------|---------------------|---------------------|---------------------|
> > |                    | clipiqa↑            | niqe↓               | musiq↑              | clipiqa↑            | niqe↓               | musiq↑              |
> > | DirectInference    | 0.522               | 4.167               | 53.98               | 0.469               | 4.370               | 29.00               |
> > | AttnEntropy        | 0.547               | 4.210               | 54.87               | 0.528               | 4.614               | 27.98               |
> > | ScaleCrafter       | 0.664               | 3.577               | 61.12               | 0.618               | 3.783               | 36.00               |
> > | FouriScale         | 0.662               | 3.580               | 60.77               | 0.612               | 3.791               | 35.52               |
> > | HiDiffusion        | **0.690**           | 4.049               | 61.69               | 0.574               | 7.348               | 36.71               |
> > | AccDiffusion       | 0.627               | 3.641               | 57.02               | 0.626               | 3.587               | 31.83               |
> > | DemoFusion         | 0.651               | 3.410               | 58.98               | 0.637               | 3.376               | 33.46               |
> > | **Ours**           | 0.668               | **3.391**           | **63.10**           | **0.646**           | **3.367**           | **37.33**           |
> >
> > We have added those results in the updated supplementary file.
> >
> >
> > **Q5 More results in real-world application scenarios.**
> >
> > Thanks for the suggestion. As suggested, to further validate the performance of FreCaS in real-world application scenarios, we have provided additional visual results in three categories:
> >
> > + **Simple scenes.** These images typically contain a single object in a realistic style. We display images of people, animals, landscapes, buildings, and other common objects. The visual results for this group are presented in Fig. 4 of the updated supplementary file.
> > + **Various styles.** This group showcases images in different artistic styles, including oil painting, pencil sketch, ink wash, watercolor, and poster art. The results are shown in the first two rows of Fig. 5 of the updated supplementary file. .
> > + **Complex scenes.** These images contain multiple objects or have intricate textures. The results are displayed in the bottom two rows of are presented in Fig. 5 of the updated supplementary file.
> >
> > From these visual results, it is evident that FreCaS consistently generates high-quality images across various styles and contents, demonstrating FreCaS' capability in real-world applications.

---

> > > ### Author Response · Authors · 2024-11-23
> > >
> > > **Q6 Computational resource demand.**
> > >
> > > Thanks for the comment. The computational resource demand primarily comes
> > >  from two aspects:
> > >
> > > **First**, the nature of high-resolution image generation using diffusion-based methods is inherently computationally intensive. The iterative process of diffusion models, combined with the need for higher resolution, significantly increases the computational requirements. Nonetheless, our FreCaS method has made significant strides in reducing computational cost. For example, FreCaS is about 7$\times$ faster than DemoFusion (the second-performing model) when generating 4096$\times$4096 images using a pre-trained SDXL model, while achieving comparable or even better image quality.
> > >
> > > **Second**, to ensure a fair comparison with other methods, we do not incorporate additional diffusion acceleration techniques, such as advanced samplers or feature utilization techniques. Integrating these strategies could further reduce the computational cost of FreCaS. For example, we can further reduce FreCaS' latency from 13 seconds to 6 seconds when integrated with the DeepCache technique on $\times 4$ image generation using SDXL.
> > >
> > > We hope our explanation can address this reviewer's concern on computational costs.

---

> > > > ### Author Response · Authors · 2024-11-25
> > > > **Your feedback and further comments are appreciated.**
> > > >
> > > > Dear Reviewer Sncf,
> > > >
> > > > We sincerely thank you for your time in reviewing our paper and your constructive comments. We have posted our point-to-point responses in the review system. Since the public discussion phase will end very soon, we appreciate if you could read our responses and let us know your feedback and further comments.
> > > >
> > > > Best regards,
> > > > Authors of paper #1039

---

> > > > > ### Author Response · Authors · 2024-12-02
> > > > >
> > > > > Dear Reviewer Sncf,
> > > > >
> > > > > Thank you very much for your time in reviewing our paper and your constructive comments. The public discussion period will end very soon by Dec 2. We are looking forward to receiving your feedback on our paper and responses.
> > > > >
> > > > > Best regards,
> > > > > Authors of paper #1039

---

### Official Review · Reviewer_N5Ek · 2024-11-03

**Soundness:** 3
**Presentation:** 3
**Contribution:** 3
**Rating:** 6
**Confidence:** 4

**Summary:**

This paper introduces an efficient frequency-aware cascaded sampling framework designed for tuning-free higher-resolution generation. The framework breaks down the sampling process into multiple stages, gradually increasing resolution at each stage. Additionally, it presents a frequency-aware classifier-free guidance approach that applies varying guidance strengths to different frequency components.
Experimental results demonstrate the effectiveness and efficiency of proposed method over various settings.

**Strengths:**

* This paper proposes a higher-resolution generation method that achieves better performance and faster sampling speeds.
* The paper is well-written, with a clear and straightforward motivation.

**Weaknesses:**

* Some evaluation metrics are missing. The authors only provided FID_b and FID_p, but did not include the FID score between the generated and real image sets, which is the most widely used metric for this task.

* Most of the 4K comparison results are in non-realistic styles, such as painting styles. More comparisons in realistic styles need to be provided.

* The cascade pipeline for high-resolution generation has been utilized in several previous works, such as DemoFusion and FouriScale, making the contribution here relatively marginal as it primarily builds on this existing approach. Therefore, a comprehensive evaluation is necessary to demonstrate the improvement brought by the addition of fa-cfg and the F step calculation.

**Questions:**

Please refer to weakness part.

---

> ### Author Response · Authors · 2024-11-23
>
> We sincerely thank this reviewer for the recognition on the effectiveness of our work. We have performed more experiments and comparisons based on this reviewer's suggestion. Please find our point-to-point responses in the following.
>
> **Q1 Missing FID scores.**
>
> Thanks for the suggestion and we are sorry for not presenting the FID scores in the submission. We have calculated the FID metrics in both the main experiments and the ablation studies. Please kindly refer to our responses to the Q1/Q2 of Reviewer z3AD for the details.
>
> **Q2 More 4K comparisons in realistic styles.**
>
> Thanks for the suggestion. As suggested, we have provided more 4K visual comparisons under realistic scenes in Fig. 7 of the updated supplementary file. As can be seen, our FreCaS consistently delivers better results in both image layout and semantic details. In addition, please refer to our responses to Q5 of reviewer Sncf for more visual results under real-world scenarios.
>
> **Q3 Differences from DemoFusion and FouriScale.**
>
> Thanks for the comment. We'd like to clarify that our FreCaS differs significantly from DemoFusion and FouriScale:
>
> + **DemoFusion** uses a cascaded sampling framework that performs multiple full diffusion processes at different resolutions, leading to repetitive computations and significant computational overhead. For example, when generating a 4096$\times$4096 image, DemoFusion needs to perform multiple full diffusion processes at lower resolutions. In contrast, FreCaS conducts a single diffusion process across resolutions. This new cascaded framework significantly reduces the computational overhead and achieves a 7.58$\times$ speedup when generating 4096$\times$4096 images using SDXL.
> + **FouriScale** does not use a cascaded framework. Instead, it generates higher-resolution images by explicitly enlarging the receptive field of convolutional layers, which is effective but limited to more advanced diffusion models like SD3. FreCaS, on the other hand, focuses on the general sampling process of diffusion models, so that it can be easily applied to other diffusion models. Additionally, FreCaS outperforms FouriScale in both image quality and inference speed when integrated with SDXL, highlighting the significant performance differences between the two methods.
>
> **Q4 Effectiveness of timestep shifting, FA-CFG and CA-reuse.**
>
> Thanks for the suggestion. To better verify the effectiveness of each component of FreCaS, we conducted more ablation studies on our proposed cascaded framework without timestep shifting, our proposed cascaded framework with timestep shifting, FA-CFG, and CA-reuse strategies. Please kindly note that the "timestep shifting" refers to the "F-step calculation" of this comment.
>
> Tab. 1 Ablation studies on $2048\times 2048$ generation of SDXL. "CF w/o TS" denotes our proposed cascaded framework without timestep shifting. "CF" is our proposed cascaded framework.
> | Model | CF w/o TS | CF | FA-CFG | CA-reuse | FID↓ | $FID_p$↓ | IS↑ | $IS_p$↑ | CLIP SCORE↑ | Latency (s) |
> |-------|-----------|----|--------|----------|------|------------------|-----|-----------------|-------------|-------------|
> | #1    |           |    |        |          | 39.14| 29.71            | 11.52| 14.60            | 32.51       | 34.10       |
> | #2    | ✓         |    |        |          | 17.50| 24.03            | 17.03| 16.34            | 33.27       | 13.70       |
> | #3    |           | ✓  |        |          | 17.62| 20.49            | 17.01| 16.54            | 33.24       | 13.71       |
> | #4    |           | ✓  | ✓      |          | 16.62| 17.91            | 17.16| 16.82            | 33.34       | 13.74       |
> | #5    |           | ✓  | ✓      | ✓        | 16.48| 17.91            | 17.18| 17.31            | 33.28       | 13.84       |
>
> Our cascaded framework significantly outperforms the baseline, with a decrease of 22.52 in the FID score and a reduction of 20.39 seconds in latency. This demonstrates the high efficiency of the new cascaded framework.
> When removing the timestep shifting, the results improve slightly on the FID and IS metrics but at the cost of a significant decrease in $\textbf{FID}_p$ and $\textbf{IS}_p$. This is because setting "F" equal to "L" shortens the diffusion process, which helps maintain the overall structure but limits the enhancement of details in expanded frequency bands.
> Our FA-CFG strategy improves both FID and IS scores and shows substantial improvement in $\text{FID}_p$, demonstrating its effectiveness in generating realistic image details.
> CA-reuse strategy further enhances $\text{IS}_p$,  indicating its effectiveness in improving semantic appearance.
> Moreover, these strategies introduce minimal additional latency.
>
> We hope our response can address this reviewer's concerns.

---

> > ### Author Response · Authors · 2024-11-25
> > **Your feedback and further comments are appreciated.**
> >
> > Dear Reviewer N5Ek,
> >
> > We sincerely thank you for your time in reviewing our paper and your constructive comments. We have posted our point-to-point responses in the review system. Since the public discussion phase will end very soon, we appreciate if you could read our responses and let us know your feedback and further comments.
> >
> > Best regards,
> > Authors of paper #1039

---

> > > ### Comment · Reviewer_N5Ek · 2024-12-01
> > > **Response to author**
> > >
> > > Thank you for your rebuttal. The response addressed most of my concerns, and I have raised my score to 6. However, I still have some reservations regarding the comparison with SR as post-processing, as mentioned by Reviewer 27vZ. It seems that most metrics do not outperform SR-based methods. Could you clarify further on the significance of the higher-resolution generation task?

---

> > > > ### Author Response · Authors · 2024-12-01
> > > >
> > > > We sincerely thank this reviewer for the valuable feedback and raising the score! Regarding the comparison with SR-based post-processing methods, please find what below our explanations.
> > > >
> > > > **Q5 Comparison with SR-based methods.**
> > > >
> > > > SR-based post-processing is a straightforward way to generate higher-resolution images. While it can serve as a good baseline, SR-based post-processing has inherent limitations in synthesizing richer details. Our detailed explanations are as follows:
> > > >
> > > > + **On the evaluation metrics.** As we explained in the responses to Reviewer 27vZ and indicated by previous works (DemoFusion and HiDiffusion), although FID, IS and CLIP Score are widely adopted for evaluating image generation results, they are problematic in assessing high-resolution image quality because they need to downsample the generated image to compute the scores. Since the SR methods are designed to strictly adhere to low-resolution inputs, this will result in higher scores of these metrics. To address this issue, researchers have proposed to evaluate high-resolution images in a cropped patch manner to better assess the details, leading to the updated metrics of $\text{FID}_p(\downarrow)$ and $\text{IS}_p(\uparrow)$. Our FreCaS achieves better results on $\text{FID}_p(\downarrow)$ and $\text{IS}_p(\uparrow)$ compared to SDXL+ESRGAN (17.91 vs. 18.10 on $\text{FID}_p$ and 17.31 vs. 16.58 on $\text{IS}_p$ for $\times4$ generations) and has similar performance to SDXL+SUPIR (17.91 vs. 17.31 on $\text{FID}_p$ and 17.31 vs. 17.12 on $\text{IS}_p$ for $\times4$ generations).
> > > >
> > > > + **Richer detail generation.** The essential limitation of SR-based post-processing for higher-resolution generation lies in that they are difficult to generate richer details. This is because the SR model is trained to adhere to the low-resolution input, and hence not too much new details will be generated in the zooming process. However, FreCaS does not have such a restriction in higher-resolution image synthesis, and it can generate much more new details, resulting in better image quality. This is why FreCaS achieves better performance on $\text{FID}_p$ and $\text{IS}_p$ metrics. We also present visual comparisons in Figure 2 of the revised appendix file. One can observe that SDXL+ESRGAN tends to simply enlarge the flowers or hairs but fails to generate rich details, whereas our FreCaS generates more vivid textures. This observation is consistent with the findings in the DemoFusion and HiDiffusion papers.
> > > >
> > > > + **Higher efficiency.** Compared with recent generative SR methods (such as SDXL-based SUPIR), our FreCaS demonstrates significant advantages in efficiency. The SDXL based SR methods can produce more details than GAN-based SR methods but they have significantly longer latency. As shown in Table 2, our FreCaS achieves comparable $\text{FID}_p$ and $\text{IS}_p$ metrics with SUPIR but with much shorter latency (13.8 seconds for FreCaS vs. 105.5 seconds for SDXL+SUPIR on $\times4$ image generations).
> > > >
> > > > Tab. 2 Comparison with super-resolution based methods on ×4 and ×16 generation of SDXL.
> > > > |   | Methods       | FID↓ | FID_p↓ | IS↑ | IS_p↑ | CLIP SCORE↑ | Latency(s)↓ |
> > > > |---|---------------|------|--------|-----|-------|-------------|-------------|
> > > > | **×4** | SDXL+ESRGAN  | 13.03 | 18.10 | 17.30 | 16.58 | 34.13 | 6.36 |
> > > > | **×4** | SDXL+SUPIR   | 12.08 | 17.31 | 17.57 | 17.12 | 34.16 | 105.5 |
> > > > | **×4** | **Ours**     | 16.48 | 17.91 | 17.18 | 17.31 | 33.28 | 13.84 |
> > > > | **×16** | SDXL+ESRGAN | 45.86 | 43.10 | 12.94 | 13.48 | 33.44 | 7.25 |
> > > > | **×16** | SDXL+SUPIR  | 43.94 | 39.35 | 13.22 | 14.37 | 33.49 | 512.4 |
> > > > | **×16** | **Ours**    | 42.75 | 39.82 | 12.68 | 14.16 | 33.03 | 85.87 |
> > > >
> > > > In addition, SR-based methods typically require much costs for preparing the training dataset (e.g., SUPIR spends significant efforts in collecting and screening the large-scale training data), whereas our FreCaS is training-free.
> > > >
> > > > We sincerely hope our explanation can address this reviewer's concerns.

---

### Official Review · Reviewer_z3AD · 2024-11-03

**Soundness:** 3
**Presentation:** 3
**Contribution:** 2
**Rating:** 6
**Confidence:** 3

**Summary:**

The paper proposes an efficient frequency-aware cascaded sampling framework designed to tackle the high computational cost of high-resolution image generation. It breaks down the sampling process into progressive stages, expanding frequency bands and refining details step-by-step. Using frequency-aware classifier-free guidance (FA-CFG) and cross-stage attention fusion, FreCaS improves both image quality and generation speed, outperforming state-of-the-art methods like ScaleCrafter and DemoFusion in generating high-resolution images.

**Strengths:**

1. Assigns different guidance strengths to components of different frequencies to achieve more finegrained control.
2. Good effiency compared with existing works.

**Weaknesses:**

1. Missing FID which is an important metric and widely adopted by exsiting works. The lacking of FID makes it difficult to comprehensively judge its effectiveness.
2. The cascade framework is widely adopted by the exisitng works. It is also necessary to introduce FID into the ablation study to validate the effectiveness of the main contributions of this work.
3. Some improvements are marginal, e.g., the results on SDXL.

**Questions:**

See the weekness part.

---

> ### Author Response · Authors · 2024-11-23
>
> We sincerely thank this reviewer for the recognition on the efficiency of our work.
>
> **Q1/Q2 Missing FID scores in the main experiments and ablation study.**
>
> Thanks for the suggestion and we are sorry for the missing of FID scores. We further evaluate all the methods and each component of our method using the FID metric.
>
> Table 1 presents the FID results of all competing methods. In this table, "DO" means "duplicated object", which indicates whether the method takes the duplicated object problem into consideration. "SpeedUP" denotes the efficiency speed-up over the DirectInference baseline. We can see that our FreCaS achieves the best FID scores in all the experiments, which aligns with the trends observed in other image quality metrics.
>
> Table 2 presents the FID results of ablation studies to validate each component of FreCaS. One can see that our cascaded framework significantly improves the performance and efficiency over the baseline, reducing the FID score by 22.52 and decreasing the latency by 20.39 seconds. The high efficiency of our cascaded framework is evident, as it performs only one diffusion process across resolutions. However, existing methods often conduct an independent diffusion process at each resolution.
>
> Additionally, our FA-CFG and CA-reuse strategies further improve the FID and IS scores, and show substantial improvements in $\text{FID}_p$ and $\text{IS}_p$, demonstrating their effectiveness in enhancing image details and semantic appearances. These strategies introduce minimal additional latency.
>
> We have added the results of FID scores in the revised manuscript.

---

> > ### Author Response · Authors · 2024-11-23
> >
> > Tab. 1 Experiments on $\times 4$ and $\times 16$ generation of SD2.1 and SDXL. The **bold** and *italic* indicate the best and second one among all methods that consider the duplicated object problem.
> > |                      |                      | Methods            | DO | FID | FID_b↓ | FID_p↓ | IS↑ | IS_p↑ | CLIP SCORE↑ | Latency(s)↓ | SpeedUP↑ |
> > |----------------------|----------------------|--------------------|----|-----|--------|--------|-----|-------|-------------|-------------|----------|
> > | **SD2.1**            | **×4**               | DirectInference    | ✕  | 31.07 | 34.54 | 23.84 | 15.00 | 17.26 | 32.01       | 5.50        | 1x       |
> > |                      |                      | MultiDiffusion     | ✕  | 21.05 | 22.44 | 14.68 | 17.46 | 18.29 | 32.49       | 120.21      | 0.046x   |
> > |                      |                      | AttnEntropy        | ✓  | 28.33 | 30.63 | *21.34* | 15.67 | **17.71** | 32.28 | 5.56 | 0.99x |
> > |                      |                      | ScaleCrafter       | ✓  | *16.65* | *13.18* | 22.44 | *17.42* | *16.29* | *32.88* | 6.36 | 0.86x |
> > |                      |                      | FouriScale         | ✓  | 19.01 | 15.33 | 23.26 | 17.11 | 15.57 | **32.92** | 11.06 | 0.50x |
> > |                      |                      | HiDiffusion        | ✓  | 19.95 | 16.21 | 25.26 | 17.13 | 16.12 | 32.37 | *3.57* | *1.54x* |
> > |                      |                      | **Ours**           | ✓  | **16.38** | **13.14** | **21.23** | **17.55** | 16.04 | 32.33 | **2.56** | **2.16x** |
> > | **SD2.1**            | **×16**              | DirectInference    | ✕  | 124.5 | 128.3 | 50.23 | 8.84 | 15.30 | 27.67       | 49.27       | 1x       |
> > |                      |                      | MultiDiffusion     | ✕  | 67.44 | 74.15 | 15.28 | 8.75 | 18.82 | 31.14       | 926.33      | 0.05x    |
> > |                      |                      | AttnEntropy        | ✓  | 122.6 | 127.6 | *46.52* | 9.31 | **16.25** | 28.33 | 49.33 | 1.00x |
> > |                      |                      | ScaleCrafter       | ✓  | 34.47 | 34.55 | 57.47 | 13.02 | 12.12 | 31.44 | 92.86 | 0.53x |
> > |                      |                      | FouriScale         | ✓  | 34.17 | *34.13* | 58.01 | 12.79 | 13.15 | *31.68* | 90.13 | 0.55x |
> > |                      |                      | HiDiffusion        | ✓  | *33.15* | 34.17 | 70.58 | *13.49* | 11.87 | 31.09 | *18.22* | *2.70x* |
> > |                      |                      | **Ours**           | ✓  | **19.95** | **20.11** | **43.71** | **15.22** | *13.74* | **31.92** | **13.35** | **3.69x** |
> > | **SDXL**             | **×4**               | DirectInference    | ✕  | 39.15 | 43.83 | 29.71 | 11.52 | 14.60 | 32.51       | 34.10       | 1x       |
> > |                      |                      | AttnEntropy        | ✓  | 36.54 | 41.30 | 27.67 | 11.69 | 15.04 | 32.71 | 34.36 | 0.99x |
> > |                      |                      | ScaleCrafter       | ✓  | 22.76 | 24.23 | 23.17 | 14.10 | 14.97 | 32.70 | 39.64 | 0.86x |
> > |                      |                      | FouriScale         | ✓  | 26.44 | 26.88 | 27.24 | 13.97 | 14.44 | 32.90 | 66.18 | 0.52x |
> > |                      |                      | HiDiffusion        | ✓  | 21.67 | 20.69 | 21.80 | 15.56 | 15.93 | 32.62 | *18.38* | *1.86x* |
> > |                      |                      | AccDiffusion       | ✓  | 19.87 | 17.62 | 21.11 | 17.07 | 16.15 | 32.66 | 102.46 | 0.33x |
> > |                      |                      | DemoFusion         | ✓  | *18.77* | *16.33* | *18.77* | *17.10* | *17.21* | *33.16* | 83.95 | 0.41x |
> > |                      |                      | **Ours**           | ✓  | **16.48** | **12.63** | **17.91** | **17.18** | **17.31** | **33.28** | **13.84** | **2.46x** |
> > | **SDXL**             | **×16**              | DirectInference    | ✕  | 145.4 | 151.3 | 62.39 | 6.41 | 11.66 | 28.24       | 312.36      | 1x       |
> > |                      |                      | AttnEntropy        | ✓  | 142.1 | 148.9 | 60.54 | 6.46 | 12.44 | 28.46 | 312.46 | 1.00x |
> > |                      |                      | ScaleCrafter       | ✓  | 71.49 | 75.11 | 73.21 | 8.68 | 9.81 | 30.76 | 560.91 | 0.56x |
> > |                      |                      | FouriScale         | ✓  | 98.01 | 77.63 | 84.05 | 8.00 | 9.41 | 30.78 | 534.08 | 0.58x |
> > |                      |                      | HiDiffusion        | ✓  | 81.48 | 83.41 | 120.1 | 9.79 | 9.56 | 29.18 | *101.59* | *3.07x* |
> > |                      |                      | AccDiffusion       | ✓  | 50.47 | 48.15 | 46.07 | 12.11 | 11.75 | 32.26 | 763.23 | 0.41x |
> > |                      |                      | DemoFusion         | ✓  | *47.80* | *44.54* | **35.52** | *12.38* | *13.82* | **33.03** | 649.25 | 0.48x |
> > |                      |                      | **Ours**           | ✓  | **42.75** | **40.63** | *39.82* | **12.68** | **14.16** | **33.03** | **85.87** | **3.64x** |

---

> > > ### Author Response · Authors · 2024-11-23
> > >
> > > Tab. 2 Ablation studies on $2048\times 2048$ generation of SDXL.
> > > | Model | Cascaded Framework | FA-CFG | CA-reuse | FID↓ | FID_p↓ | IS↑ | IS_p↑ | CLIP SCORE↑ | Latency (s) |
> > > |-------|--------------------|--------|----------|------|--------|-----|-------|-------------|-------------|
> > > | #1    |                    |        |          | 39.14 | 29.71  | 11.52 | 14.60 | 32.51       | 34.10       |
> > > | #2    | ✓                  |        |          | 17.62 | 20.49  | 17.01 | 16.54 | 33.24       | 13.71       |
> > > | #3    | ✓                  | ✓      |          | 16.62 | 17.91  | 17.16 | 16.82 | 33.34       | 13.74       |
> > > | #4    | ✓                  | ✓      | ✓        | 16.48 | 17.91  | 17.18 | 17.31 | 33.28       | 13.84       |
> > >
> > >
> > > **Q3 Marginal improvements on SDXL.**
> > >
> > > We'd like to clarify that our improvement is not marginal when using pretrained SDXL models. For example, as shown in Table 1,
> > > DemoFusion achieves higher FID scores over AccDiffusion by only 1.10 and 2.67 for 2048$\times$2048 and 4096$\times$4096 image generations, respectively. In contrast, our FreCaS improves the FID scores over DemoFusion by 2.29 and 5.05, respectively, while it is 6.0$\times$ and 7.58$\times$ faster than DemoFusion. These improvements are not trivial compared with the existing training-free based methods.

---

> > > > ### Author Response · Authors · 2024-11-25
> > > > **Your feedback and further comments are appreciated.**
> > > >
> > > > Dear Reviewer z3AD,
> > > >
> > > > We sincerely thank you for your time in reviewing our paper and your constructive comments. We have posted our point-to-point responses in the review system. Since the public discussion phase will end very soon, we appreciate if you could read our responses and let us know your feedback and further comments.
> > > >
> > > > Best regards,
> > > > Authors of paper #1039

---

> > > > > ### Author Response · Authors · 2024-12-02
> > > > >
> > > > > Dear Reviewer z3AD,
> > > > >
> > > > > Thank you very much for your time in reviewing our paper and your constructive comments. The public discussion period will end very soon by Dec 2. We are looking forward to receiving your feedback on our paper and responses.
> > > > >
> > > > > Best regards,
> > > > > Authors of paper #1039

---

### Official Review · Reviewer_27vZ · 2024-11-03

**Soundness:** 2
**Presentation:** 3
**Contribution:** 2
**Rating:** 5
**Confidence:** 5

**Summary:**

This paper introduces FreCaS, a training-free approach for high-resolution image generation based on a diffusion model. FreCaS progressively upscales images using Frequency-Aligned Classifier-Free Guidance, which guides the diffusion model to incorporate new details within the expanded frequency domain. Experimental results show that FreCaS outperforms existing training-free methods in both efficiency and performance. Furthermore, the approach can be seamlessly integrated with advanced models, such as SD3.

**Strengths:**

1. The proposed method is simple and useful. FreCaS leverages the latent space of lower resolutions, thereby reducing computational costs in the initial stages of higher-resolution generation. By efficiently reusing low-frequency information from lower-resolution outputs, it streamlines the generation process. This simplicity and effectiveness make FreCaS adaptable to other diffusion-based text-to-image models.
2. The ablation study is extensive and thorough, covering critical factors such as inference steps at each resolution stage, adjustments in CFG weights, and reuse of cross-attention maps.

**Weaknesses:**

1. The coarse-to-fine approach and frequency domain enhancement are commonly applied in training-free high-resolution image generation methods, such as the latest HiPrompt and ResMaster. Although these methods do not use CFG, combining such techniques in this paper does not provide particularly new insights.
2. Generating higher-resolution images can be approached through various methods. This paper focuses primarily on training-free comparisons, but including comparisons with training-based methods, such as Pixart-Sigma or UltraPixel, or super-resolution methods like ESRGAN and SUPIR, would provide a more comprehensive analysis.
3. Training-free methods typically yield low success rates, often requiring tens of attempts to obtain a single satisfactory result. I am not convinced that training-free methods offer a viable solution for high-resolution image generation, especially given that training-based methods are generally faster, more reliable, and stable. Additionally, some training-based methods do not require significant computational resources.

**Questions:**

1. In Figure 6, the weight of the cross-attention map appears to influence image details and textures (e.g., the blueberry's surface) rather than the layout, which remains stable as the weight varies from 0 to 1. This observation seems to contradict the claim in Section 3.4, which states that cross-attention maps capture the semantic layout by representing attention weights from interactions between spatial features and textual embeddings. Could you clarify whether cross-attention maps primarily control fine-grained details and textures, or if they indeed play a more critical role in determining the semantic layout as suggested earlier?
2. What are the GPU memory requirements for each of the compared methods and for the proposed FreCaS method?

---

> ### Author Response · Authors · 2024-11-23
>
> We sincerely thank this reviewer for the recognition on the effectiveness of our work. We have conducted extensive additional experiments based on this reviewer's constructive suggestions. Please find what below our point-to-point responses to this reviewer's comments.
>
> **Q1 New insights of FreCaS.**
>
> Thanks for your constructive comments. While the coarse-to-fine and frequency domain enhancement approaches have been used in many existing works, we would like to clarify that our FreCaS is not a simple combination of existing coarse-to-fine approaches and frequency-domain operations, but it has significant differences from existing methods such as HiPrompt and ResMaster. In specific, FreCaS introduces several novel insights in *efficient  coarse-to-fine design* and *details enhancement in frequency domain*.
>
> **First**, FreCaS presents a novel *efficient coarse-to-fine sampling framework*. Current methods, such as DemoFusion and ResMaster, perform a complete diffusion process at each resolution. This leads to redundant computations in the early stages of high-resolution diffusion, as much of the information is already obtained in the low-resolution diffusion process.
> In contrast, FreCaS transitions the diffusion from low to high resolutions in just one process. Such a framework significantly reduces the computational overhead. For example, while ResMaster is about twice as fast as DemoFusion in generating 4096$\times$4096 images using SDXL, FreCaS achieves a 7.58$\times$ speedup, which is significantly faster while keeping the generation quality.
>
> **Second**, our Frequency-Aware Classifier-Free Guidance (FA-CFG) strategy efficiently and effectively enhances details in the frequency domain. Existing methods, such as HiPrompt and ResMaster, focus on maintaining semantic consistency by aligning low-frequency components with global prompts or low-resolution reference. In contrast, FreCaS emphasizes on generating more details by assigning higher CFG strengths to high-frequency components. Our FA-CFG strategy is different yet complementary to HiPrompt and ResMaster, potentially offering additional performance benefits when used in conjunction with these methods. Unfortunately, HiPrompt and ResMaster do not release their codes for reproduction.
>
> Additionally, FA-CFG allows for much higher CFG strengths (from 7.5 to 35) without introducing artifacts, which is a key challenge for using CFG strategy in image generation.
>
> We will add the above explanations in the related work section of the revised manuscript. Hope our clarifications can address this reviewer's concerns.

---

> > ### Author Response · Authors · 2024-11-23
> >
> > **Q2 Comparison with training-based / SR-based methods.**
> >
> > Thanks for your suggestions. As suggested, we conducted additional experiments comparing FreCaS with training-based methods (Pixart-Sigma and UltraPixel) and super-resolution methods (ESRGAN and SUPIR). To ensure fair comparisons, we set the model precision to fp16 (bf16 for UltraPixel, as recommended by the authors) and use the DDIM sampler for diffusion-based methods. For Pixart-Sigma, we can only report its results for 2048$\times$2048 image generation since its 4K model is not available. The quantitative results are summarized in Table 1.
> >
> > Tab. 1 Comparison with training-based and super-resolution based methods on $\times 4$ and $\times 16$ generation of SDXL.
> > |       | Methods         | FID↓   | $\text{FID}_p$↓ | IS↑   | $\text{IS}_p$↑ | CLIP SCORE↑ | Latency(s)↓ |
> > |-------|-----------------|--------|--------|-------|-------|-------------|-------------|
> > | **×4**| Pixart-Sigma    | 26.11  | 38.58  | 14.44 | 14.45 | 28.10       | 71.45       |
> > |       | UltraPixel      | 25.56  | 19.95  | 17.11 | 17.10 | 33.17       | 41.70       |
> > |       | SDXL+ESRGAN     | 13.03  | 18.10  | 17.30 | 16.58 | 34.13       | 6.36        |
> > |       | SDXL+SUPIR      | 12.08  | 17.31  | 17.57 | 17.12 | 34.16       | 105.5       |
> > |       | **Ours**        | 16.48  | 17.91  | 17.18 | 17.31 | 33.28       | 13.84       |
> > | **×16**| UltraPixel     | 51.43  | 45.88  | 12.48 | 13.73 | 33.07       | 162.4       |
> > |       | SDXL+ESRGAN     | 45.86  | 43.10  | 12.94 | 13.48 | 33.44       | 7.25        |
> > |       | SDXL+SUPIR      | 43.94  | 39.35  | 13.22 | 14.37 | 33.49       | 512.4       |
> > |       | **Ours**        | 42.75  | 39.82  | 12.68 | 14.16 | 33.03       | 85.87       |
> > ||
> >
> > From Table 1, we can see that FreCaS outperforms Pixart-Sigma and UltraPixel in most metrics. For example, FreCaS achieves an FID score of 16.48 and an IS score of 17.18, compared to 26.11 and 14.44 of Pixart-Sigma, and 25.56 and 17.11 of UltraPixel on the $\times 4$ image generation. This is because Pixart-Sigma, as acknowledged by the authors, heavily relies on the advanced samplers (see [link](https://github.com/PixArt-alpha/PixArt-sigma/issues/65)) so that the results are not very stable. UltraPixel, while achieving comparable performance to DemoFusion, still lags behind FreCaS in most metrics. Besides, the two methods are much slower than our FreCaS.
> >
> > For SR-based methods, FreCaS may have lower FID, IS, and CLIP scores than SDXL+ESRGAN. This is because SR methods are designed to strictly adhere to low-resolution inputs, while these metrics (FID, IS, and CLIP) evaluate images by downsampling them to low resolution, which cannot well reflect the quality of generated high-resolution images. FreCaS significantly outperforms SDXL+ESRGAN in $\text{FID}_p$ and $\text{IS}_p$. Specifically, FreCaS achieves an $\text{FID}_p$ score of 39.82 and an $\text{IS}_p$ score of 14.16, compared to 43.10 and 13.48 of SDXL+ESRGAN on $\times 16$ image generation.
> > This indicates its superior ability to generate high-resolution local details. This observation is consistent with the findings in the DemoFusion paper. Additionally, SDXL+SUPIR outperforms FreCaS in $\text{FID}_p$ and $\text{IS}_p$, but at the cost of much longer inference latency (85.87 seconds for FreCaS vs. 512.4 seconds for SDXL+SUPIR on $\times 16$ image generations).
> >
> > We have provided some visual comparisons in Fig. 2 of the updated supplementary file. One can see that FreCaS demonstrates better visual quality than either training-based or SR-based methods in high-resolution image generation.

---

> > > ### Author Response · Authors · 2024-11-23
> > >
> > > **Q3 The merits of training-free methods.**
> > >
> > > We fully understand this reviewer's concerns regarding the viability of training-free methods for high-resolution image generation. We agree that training-based methods have distinctive advantages yet they also have several drawbacks, while training-free methods can provide a compelling alternative, especially for high-resolution image generation. We believe that training-based and training-free methods are not contradictory yet they are complementary. We explain from the following perspectives:
> > >
> > >
> > > + **Data Requirements.** Training-based methods typically require large datasets, which can be a significant challenge for high-resolution image generation. For instance, while ImageNet (1.28 million images) is sufficient for 256$\times$256 image generation, Laion5B (5.85 billion images) is necessary for 512$\times$512 and 1024$\times$1024 generations. As resolution increases, both the quantity and quality of images become a bottleneck, making it difficult to gather enough training data. In contrast, training-free methods can generate high-resolution images without the need for extensive datasets.
> > >
> > > + **Computational Resources.** High-resolution image generation with training-based methods require substantial computational resources due to the high-dimensional nature of the task. Techniques to reduce computational costs often compromise performance. For example, UltraPixel, a training-based method, requires eight A100 GPUs for training, while most training-free methods can be optimized using a single A100 GPU.
> > >
> > > + **Generalization.** Training-based methods may struggle with generating images whose settings are not seen during training. For example, LDM fails to generate images with aspect ratios other than 1:1, and SD3 cannot directly generate resolutions not encountered during training. In contrast, training-free methods like our FreCaS can handle such scenarios more effectively.
> > >
> > > + **Efficiency.** Training-free methods can be faster. For example, our FreCaS achieves faster inference speeds and better performance compared to PixArt-Sigma and UltraPixel (see Table 1).
> > >
> > > + **Complementarity.** Training-based and training-free methods are not contradictory but complementary. For instance, SDXL, a prominent training-based method for 1024$\times$1024 image generation, has been enhanced by several training-free strategies (HiDiffusion, DemoFusion, AccDiffusion, and our FreCaS), improving its performance and generation capabilities.
> > >
> > >
> > > We sincerely hope our explanations above can address this reviewer's concern on the viability of training-free methods in image generation.
> > >
> > >
> > > **Q4 Clarification on Cross-Attention Maps.**
> > >
> > > Thanks for your comment. We are sorry for any confusion caused. Actually, the observation is not contradictory to our claim. The cross-attention maps primarily control the semantic layout of the entire image, and their influence on fine-grained details and textures is a natural consequence of the strong semantic guidance provided by the cross-attention maps.
> > >
> > > In Figure 6, when the re-utilization weight is set to 0, the result shows a blueberry with strawberry textures. This occurs because in the initial low-resolution stage, the object is identified as a blueberry, while during the high-resolution diffusion process, the object is misidentified as a strawberry, leading to the addition of strawberry textures. By increasing the re-utilization weight, we strengthen the blueberry semantics at the corresponding positions, and hence a blueberry is  accurately presented in the final results.

---

> > > > ### Author Response · Authors · 2024-11-23
> > > >
> > > > **Q5 GPU memory.**
> > > >
> > > > Thanks for the suggestion.
> > > > We further report the GPU memory usage of all methods in Table 2 below. As can be seen, most methods (including our FreCaS) consume a similar amount of GPU memory: approximately 4.8 GB, 11.9 GB, 21.75 GB, and 11.75 GB for the 4$\times$ and 16$\times$ generation of SD2.1 and SDXL, respectively. This is because the VAE decoder dominates the GPU memory usage, and all methods use the same decoder.
> > > >
> > > > Notably, when generating 4096$\times$4096 images with SDXL, all methods use the decoder tiling strategy to avoid GPU memory exhaustion, resulting in lower memory usage compared to generating 2048$\times$2048 images with the same model.
> > > >
> > > > Tab. 2 The GPU memory usage on $\times 4$ and $\times 16$ generation of SD2.1 and SDXL on an A100 GPU. The results with ``*" indicates inference with decoder tiling strategy.
> > > >
> > > > | Methods            | SD2.1              | SD2.1              | SDXL               | SDXL               |
> > > > |--------------------|--------------------|--------------------|--------------------|--------------------|
> > > > |                    | ×4                 | ×16                | ×4                 | ×16                |
> > > > | DirectInference    | 4.81               | 11.93              | 21.74              | 11.76*             |
> > > > | MultiDiffusion     | 4.81               | 11.93              | -                  | -                  |
> > > > | AttnEntropy        | 4.81               | 11.93              | 21.74              | 11.76*             |
> > > > | ScaleCrafter       | 4.81               | 11.94              | 21.72              | 13.30*             |
> > > > | FouriScale         | 4.81               | 11.94              | 21.72              | 24.29*             |
> > > > | HiDiffusion        | 4.80               | 11.93              | 21.72              | 11.80*             |
> > > > | DemoFusion         | -                  | -                  | 21.75              | 11.73*             |
> > > > | AccDiffusion       | -                  | -                  | 21.80              | 18.07*             |
> > > > | **Ours**           | 4.81               | 11.94              | 21.74              | 11.75*             |
> > > > ||

---

> > > > > ### Author Response · Authors · 2024-11-25
> > > > > **Your feedback and further comments are appreciated.**
> > > > >
> > > > > Dear Reviewer 27vZ,
> > > > >
> > > > > We sincerely thank you for your time in reviewing our paper and your constructive comments. We have posted our point-to-point responses in the review system. Since the public discussion phase will end very soon, we appreciate if you could read our responses and let us know your feedback and further comments.
> > > > >
> > > > > Best regards,
> > > > > Authors of paper #1039

---

> ### Comment · Reviewer_27vZ · 2024-11-27
> **Response to Authors**
>
> Thank you for the explanation. The missing details have been well supplemented. I appreciate the efficiency of the proposed method, especially when compared with other training-free methods. However, there are some concerns regarding comparisons with training-based and SR-based methods. From Table 1, it is evident that SDXL+ESRGAN or SDXL+SUPIR outperform FreCaS in many metrics, with the efficiency of SDXL+ESRGAN being particularly noteworthy. This raises two possibilities:
> 1. The evaluation metrics used may not be reasonable, despite their wide use in previous work.
> 2. The SDXL+ESRGAN pipeline is already an effective and efficient solution, negating the need for more complex methods.
>
> While I acknowledge that training-based methods may take more time to generate high-resolution results, I doubt that the results produced are inferior to those of training-free methods. Additionally, it is crucial to provide the stability and success rate of the proposed method. My experience with most training-free methods indicates that it usually takes numerous attempts to achieve a relatively good result, despite yielding favorable quantitative evaluation results.

---

> > ### Author Response · Authors · 2024-11-28
> >
> > We sincerely thank this reviewer for the further comments and the recognition on the efficiency of our proposed method. We answer this reviewer's questions from two aspects, comparison with SR-based methods and comparison with training-based methods. Please find what below our responses.
> >
> > **Q6 Comparison with SR-based methods.**
> >
> > We first address this reviewer's concerns on the comparison with SR-based methods, especially the SDXL-ESRGAN.
> >
> > + **Evaluation metrics.**
> > There are four image quality metrics used in Table 1, FID, IS, FID$_p$ and IS$_p$. Among them, SDXL+ESRGAN outperforms FreCaS in FID and IS, while FreCaS outperforms SDXL+ESRGAN in FID$_p$ and IS$_p$. As indicated by this reviewer, although FID and IS are widely adopted for evaluating image generation results, they are actually problematic in evaluating the generated image quality. The limitations of FID and IS have also been indicated in the papers of DemoFusion and HiDiffusion. Specifically, FID and IS metrics assess images at a downsampled resolution, leading to inaccurate assessment on fine details. To address this issue, FID$_p$ and IS$_p$ are defined to evaluate images in a cropped patch-based manner, which greatly reduce the negative impact of downsampling processing on image quality. We can observe that our FreCaS outperforms SDXL+ESRGAN on both $\times4$ and $\times16$ generations in terms of FID$_p$ and IS$_p$, validating the advantages of FeCaS in generated image quality.
> >
> > + **The results of SDXL+ESRGAN.** SDXL+ESRGAN can serve as a baseline for higher-resolution image generation solution but it has some drawbacks. Due to the limited generation capability of ESRGAN, the images generated by SDXL+ESRGAN typically contains fewer details. This can be observed from Table 1: SDXL+ESRGAN always lags behind FreCaS on FID$_p$ and IS$_p$ on both $\times4$ and $\times16$ generations. Besides, we also provided some visual comparisons in Fig. 2 of the revised supplementary file. One can see that the SDXL+ESRGAN tends to simply enlarge the flowers or hairs but fails to generate rich details, whereas our FreCaS generates more vivid textures.
> >
> > In summary, our FreCaS demonstrates superior performance compared to SDXL+ESRGAN, particularly in terms of fine details, which is verified by the patch-based metrics FID$_p$ and IS$_p$ as well as visual results. FreCaS is a competitive alternative to SR-based methods, capable of efficiently generating richer and clearer details.

---

> > > ### Author Response · Authors · 2024-11-28
> > >
> > > **Q7 Comparison with training-based methods.**
> > >
> > > We fully understand this reviewer’s concerns regarding the stability and success rate of training-free methods when compared to training-based methods. Due to the time limit, we are not able arrange a user study to evaluate the success rate of each method (we will conduct such a study later and add the results in the revised manuscript). To quantitatively analyze the stability of training-based and training-free methods, we generated 200 images for 20 randomly selected prompts (10 images for each prompt) using Pixart-Sigma, UltraPixel, and our FreCaS. Considering that FID and IS are not suitable for evaluating individual examples, we adopt the NR-IQA metrics (CLIPIQA, NIQE, and MUSIQ) to measure the performance of each method. In specific, we define the following three measures to evaluate the generation quality, stability and consistency of each method.
> > >
> > > + **Average Score (Mean):** The average score across the 200 generated images for each of the three metrics (CLIPIQA, NIQE, and MUSIQ). This metric can reflect the generation quality of each method.
> > >
> > > + **Average of Standard Deviations (AoS):** We first compute the standard deviation of the metrics for each prompt across 10 runs, and then report the average of these standard deviations across all 20 prompts. This metric can reflect the stability of each method.
> > >
> > > + **Standard Deviation of Averages (SoM):** We first compute the mean of the metrics for each prompt across 10 runs, and then report the standard deviation of these mean values across all 20 prompts. This metric can reflect the consistency of a method's performance across different prompts.
> > >
> > > The results are listed in Table 3.
> > >
> > > Tab. 3 Experiments on 200 images of 20 prompts on $\times4$ generation.
> > > | Methods      | clipiqa (Mean) | clipiqa (AoS) | clipiqa (SoM) | niqe (Mean) | niqe (AoS) | niqe (SoM) | musiq (Mean) | musiq (AoS) | musiq (SoM) |
> > > |--------------|----------------|---------------|---------------|-------------|------------|------------|--------------|-------------|-------------|
> > > | Pixart-Sigma | 0.558          | 0.05          | 0.11          | 5.256       | 0.35       | 0.95       | 51.546       | 4.49        | 8.24        |
> > > | UltraPixel   | 0.540          | 0.04          | 0.11          | 4.625       | 0.42       | 1.55       | 56.215       | 2.94        | 7.95        |
> > > | FreCaS       | 0.633          | 0.11          | 0.04          | 3.886       | 0.87       | 0.27       | 59.756       | 9.64        | 2.95        |
> > >
> > > From this table, we can see that our FreCaS achieves the highest "Mean" scores across the three metrics, demonstrating the best performance in term of generation quality. Pixart-Sigma and UltraPixel have smaller AoS scores than FreCaS, indicating better stability for the same input prompt. However, FreCaS demonstrates significantly better SoM scores than Pixart-Sigma and UltraPixel, indicating that it can consistently achieve better results across various prompts.
> > >
> > >
> > > Based on the above results, we argue that while FreCaS has a little lower stability compared with training-based methods, it has some distinct advantages in generation quality and consistency. In addition, from Table 1 we can see that FreCaS outperforms Pixart-Sigma and UltraPixel in all metrics under the same settings, demonstrating its superior higher-resolution generation performance. Here we further provide the CLIPIQA, NIQE, and MUSIQ metrics in Table 4, which are widely used in low-level vision tasks to assess image quality. As we can see, FreCaS achieves substantial improvements over Pixart-Sigma and UltraPixel, further demonstrating its advantages in generating better quality images. Finally, as shown in Fig. 2 of the revised supplementary file, FreCaS has better performance in generating image details, such as the flowers and hairs.
> > >
> > > Tab. 4 Non-reference IQA metrics on $\times 4$ and $\times 16$ generation of SDXL.
> > >
> > > |   | Methods      | clipiqa (↑) | niqe (↓) | musiq (↑) | Latency (s) (↓) |
> > > |---|--------------|-------------|----------|-----------|-----------------|
> > > | ×4 | Pixart-Sigma | 0.484       | 6.03     | 30.49     | 71.45           |
> > > |  | UltraPixel   | 0.587       | 4.11     | 59.13     | 41.70           |
> > > |  | **Ours**     | 0.668       | 3.39     | 63.10     | 13.84           |
> > > | ×16 | UltraPixel   | 0.510       | 4.36     | 34.68     | 162.4           |
> > > |  | **Ours**     | 0.646       | 3.37     | 37.33     | 85.87           |
> > >
> > > Actually, FreCaS is built upon the pretrained SDXL models and further enhances SDXL's generation capabilities to higher resolutions. In this sense, training-free and training-based methods are indeed complementary but not contradictory. We sincerely hope this reviewer can agree with us.

---

> > > > ### Comment · Reviewer_27vZ · 2024-11-29
> > > > **Response to Authors**
> > > >
> > > > Thank you for the additional information. Based on non-reference metrics, the proposed FreCaS outperforms training-based methods. However, I have reservations about these metrics, as they often yield high scores for images with artificial textures. Currently, human evaluation remains the most reliable method. I recommend that the authors place greater emphasis on user studies in their evaluation process. Also, apart from FID and IS, the proposed method underperforms compared to SDXL+SR methods on CLIP Score at both x4 and x16 scales. While I commend the authors’ efforts in exploring training-free methods, the available evidence suggests that their stability is limited, which may affect their practical applicability. I would like to give a borderline score.

---

> ### Author Response · Authors · 2024-11-30
>
> We thank this reviewer for spending his/her precious time continuing to discuss with us. We truly appreciate it. However, we respectfully cannot agree with this reviewer's opinion in downgrading training-free based methods. While training-based methods have advantages on stability, our training-free based FreCaS method has clear advantages on generation quality. This reviewer argued that NR-IQA metrics are not perfect to evaluate image quality, however, the FID, IS and CLIP Score metrics are not perfect either. We feel it is somewhat unfair to emphasize some metrics while ignoring other metrics to judge the performance of a method.
>
> Considering that none of the existing metrics is perfect to evaluate image generation methods, we agree with this reviewer that human subjective evaluation can be a relatively more reliable approach. Actually, we have conducted a user study in the past two days after we submitted the response to this reviewer. We just obtained the results. Please refer to Table 5 and our responses below for the details.
>
> **Q8 User studies.**
>
> We conducted a user study to explore the generated image quality and success rate of Pixart-Sigma, UltraPixel, and our FreCaS. The results are listed in the following Table 5.
>
> For the study on generation quality, we randomly select 20 prompts from Laion5B and generate one image per method for each prompt, creating 20 sets of images (3 images per set). Five volunteers (3 males and 2 females) were invited to participate in the test. All the volunteers are not working in the area of image generation to avoid potential bias. Each time, the set of 3 images for the same prompt are presented to the volunteers in random order. The volunteers can view the images multiple times, and they are asked to select the image with the best quality from each set. There are 100 votes in total.
>
> For the study on success rate, we randomly select 10 prompts and generate five images per prompt for each method, resulting in 50 images per method. We invited the same five volunteers as in the study of generation quality to judge whether the generated image is a success or failure. When making the decision, the volunteers are instructed to consider two factors. First, whether the image content is faithful to the description of the prompt. Second, whether the image quality is satisfactory. Only when both the two requirements are met, the generation is considered as a success. There are 250 judges for each method.
>
> Tab. 5 User Studies on $\times 4$ Generation with SDXL.
> | Methods      | Image Quality (Counts) | Image Quality (Percentage) | Success Rate (Counts) | Success Rate (Percentage) |
> |--------------|------------------------|----------------------------|----------------------|---------------------------|
> | Pixart-Sigma | 5                      | 5%                         | 52                   | 20.8%                     |
> | UltraPixel   | 37                     | 37%                        | 96                   | 38.4%                     |
> | **Ours**     | 58                     | 58%                        | 68                   | 27.2%                     |
>
>
> As we can see from Table 5, our FreCaS outperforms significantly Pixart-Sigma and UltraPixel in terms of image generation quality, with 58% of the images being voted as the best. In terms of success rate, UltraPixel works the best, with 96 out of 250 images being marked as successful. Our FreCaS lags behind, with 68 successful cases. However, we argue that our FreCaS still generates more successful results than Pixart-Sigma (52 images), indicating that a well designed training-free method can surpass training-based methods.
>
> From Table 5, we can also observe that none of the methods, including training-based and training-free ones, achieves a success rate higher than 40%. This implies that there are much space to improve. A hybrid approach that can take the advantages of both training-based and training-free methods can be a promising solution.

---

### Meta-Review · Area_Chair_i1uc · 2024-12-20

**Metareview:**

This paper proposes FreCaS, a training-free method to generate resolution beyond the trained resolution of a given text-to-image generation model. The reviewers generally acknowledge effectiveness and efficiency of the proposed methods, and have concerns about the stability of the proposed method. Given the mixed ratings received by reviewers, the AC carefully read the paper, review, and rebuttal, and agree that the proposed training-free method could be valuable to the community, especially for scenarios that consist of limited resources. Also, the exploration of the intrinsic properties of the generation process is also worth exploring. Therefore, the AC recommends an acceptance.

**Additional Comments On Reviewer Discussion:**

The reviewers raised useful comment in the review, but a relatively quiet in the discussion period. The AC went through the review and discussion and believe the proposed training-free paradigm could provide insights to the research community.

---

### Decision · Program_Chairs · 2025-01-22

Accept (Poster)